# The disordered p53 transactivation domain is the target of FOXO4 and the senolytic compound FOXO4-DRI

Benjamin Bourgeois[1,5], Emil Spreitzer[1,5], Daniel Platero-Rochart[1], Margret Paar[1], Qishun Zhou[1], Sinem Usluer[1], Peter L. J. de Keizer [2,3], Boudewijn M. T. Burgering [2], Pedro A. Sánchez-Murcia [1,4] & Tobias Madl [1,4] ✉

A central process contributing to the phenotype of aging is cellular senescence. We recently identified the FOXO4 – p53 axis as pivotal in maintaining the viability of senescent cells, and that senescent cells can be targeted selectively with the senolytic peptide FOXO4-DRI. Here, we solve the solution NMR structural models of the p53 transactivation domain in complex with the FOXO4 forkhead domain and in complex with FOXO4-DRI. Strikingly, we find that the disordered FOXO4-DRI binds to the disordered p53[TAD2] and forms a transiently folded complex. In this complex, both, the FOXO4-derived region and the cationic cell permeability peptide contribute to the interaction. Furthermore, we show that p53 phosphorylation enhances the affinity for both FOXO4 and FOXO4-DRI. Summarizing we provide a detailed characterization of the interaction of p53 with FOXO4 and FOXO4-DRI which is the basis for development of p53 inhibitors to treat diseases linked to cellular senescence such as cancers.

Aging is characterized by the gradual loss of physiological function over time. Multiple cellular processes are considered to contribute to this functional decline and these are collectively referred to as the hallmarks of aging and these include processes like genomic instability, telomere attrition, epigenetic alterations, loss of proteostasis, disabled macroautophagy, deregulated nutrient-sensing, mitochondrial dysfunction, cellular senescence, stem cell exhaustion, altered intercellular communication, chronic inflammation, and dysbiosis[1]. A variety of cellular stresses, including DNA damage can cause the induction of cellular senescence, which represents a multifaceted and highly heterogeneous cellular state characterized by irreversible growth arrest and importantly the secretion of a multitude of factors, including proteases and chemokines[2]. The latter is referred to as the senescence associated secretory phenotype (SASP) and this is considered to be driving many aspects of functional decline. Therefore, the clearance of senescent cells through apoptosis is one of the proposed mechanisms to reduce the burden imposed by the presence of senescent cells and in consequence, prevent age-related diseases and ultimately revert aging[3]. So-called senolytic drugs, are compounds that are effective both in vitro and in vivo, in targeting and eliminating senescent cells[4]. For example, the combination of the cancer drug Dasatinib and the natural compound Quercetin was shown to induce apoptosis in senescent cells in a variety of contexts in vitro and in vivo[1,4]. This combination treatment alleviated multiple disorders in in vivo experiments, including cardiomyocyte hypertrophy, glucose and insulin tolerance, and Alzheimer's disease, and is currently being tested in multiple clinical studies for different age-related diseases.

[1]Division of Medicinal Chemistry, Otto-Loewi Research Center, Medical University of Graz, Graz, Austria. [2]Department of Molecular Cancer Research, Center for Molecular Medicine, Division of Biomedical Genetics, University Medical Center Utrecht, Utrecht University, Utrecht, The Netherlands. [3]Cleara Biotech B.V., Yalelaan 62, 3584CM, Utrecht, The Netherlands. [4]BioTechMed-Graz, Graz, Austria. [5]These authors contributed equally: Benjamin Bourgeois, Emil Spreitzer. ✉e-mail: tobias.madl@medunigraz.at

Additionally, BCL2-inhibitors have been shown to have senolytic effects in some senescent cell models[1,4].

We recently described the cell-penetrating peptide FOXO4-DRI as a senolytic drug[5]. This compound selectively targets the interaction of p53 and Forkhead box protein O 4 (FOXO4) in senescent cells. Interference of the p53-FOXO4 interaction resulted in selective apoptosis of senescent cells and restoration of tissue homeostasis in multiple mouse models. Subsequently, beneficial effects of FOXO4-DRI have been demonstrated in various age-related models[6–8]. However, despite extensive research, the molecular details underlying the interaction of FOXO4 and p53 and the mode of action of FOXO4-DRI remain elusive.

The tumor-suppressor p53 plays a pivotal role in cellular responses to various stressors. It aids in coping with stress in various ways e.g., mediating DNA repair, cell growth arrest and apoptosis induction[9,10]. Mutations within the gene encoding p53 are among the most common mutations found in human cancers, resulting in the loss of normal transcriptional activity and acquisition of new functions that drive tumor progression[11,12]. In unstressed cells, p53 expression levels are low. However, stress stimuli, such as DNA damage or starvation, elevate p53 expression. In this context, regulation of p53 protein expression is not only achieved at the transcriptional level but also by the regulator Mdm2 which mediates p53 ubiquitination and subsequent degradation via the proteasome. Consequently, stress induction leads to p53 phosphorylation, which in turn reduces Mdm2 binding and causes p53 stabilization. The tumor-suppressive functions of p53 can either be achieved transcriptionally-dependent where p53 directly regulates target genes or transcriptionally-independent via induction of mitochondrial-dependent apoptosis[9,10,13].

p53 executes its diverse roles via distinct functional regions within its amino acid sequence. These regions include the N-terminal transactivation domain (TAD, residues 1–61), proline-rich domain (PRD, residues 62–90), DNA-binding domain (DBD, residues 91–292), a nuclear localization signal (NLS, residues 292–327), tetramerization domain (TETD, residues 327–356) and a C-terminal regulatory domain (CTD, residues 356–393; Fig. 1a)[10,13]. Several domains mediate protein-protein interactions, which are essential for fine-tuning p53 activity. Notably, the intrinsically disordered N-terminal TAD (p53$^{TAD}$) comprises two subdomains, TAD1 and TAD2. TAD1 and TAD2 play critical roles in transcriptional regulation by directly interacting with the DBD[14,15] and various (transcriptional) regulators, including CBP/p300 (CREB (cAMP-response-element-binding protein)-binding protein), high mobility group B1 (HMGB1), positive cofactor 4 (PC4), breast cancer type 2 susceptibility protein 2 (BRCA2), HDM2/MDM2, meiotic recombination 11 homolog 1 (MRE11)–DNA repair protein, and poly-PR/GR dipeptide repeats derived from the chromosome 9 open reading frame 72 (C9orf72)[16–22]. Recent studies have explored the interaction of FOXO4 and p53 and localized the interaction sites to p53's TAD and DBD and FOXO4's forkhead and C-terminal transactivation domain (CR3)[5,23,24].

FOXO4 is a member of the FOX family of transcription factors, which comprises more than 50 members distributed across 19 subclasses (FOXA to FOXS)[25,26]. A common feature among all family members is the conserved DNA binding domain called the "forkhead" (FH) or "winged-helix" domain, which is essential for transcriptional activity. FOX proteins, including FOXO4, participate in a plethora of essential biological processes, such as, but not limited to, cell cycle control, cell differentiation and proliferation, tissue homeostasis, aging, metabolism regulation and stress tolerance[27–33]. Regulation of FOXO transcriptional activity typically involves the phosphoinositide 3-kinase (PI3K) pathway and changes in the cellular redox state, often referred to as oxidative stress-mediated signaling pathways. Notably, FOXO4 levels are elevated in senescent cells, contributing to their viability during cellular senescence[5]. FOXO4 is composed of several distinct regions. The folded FH DNA-binding domain (FOXO4$^{FH}$) is surrounded by long intrinsically disordered regions. The N-terminal disordered region contains the conserved region 1 (CR1), which includes a phosphorylation site for PKB/AKT. The C-terminal disordered region harbors multiple conserved regions, namely CR$^{PKB/AKT}$, CR2, and CR3 (FOXO4$^{CR3}$). Among the regulatory mechanisms controlling FOXO4 function, the CR3 region serves as transactivation domain. Despite low sequence conservation considering the full-length proteins, the sequence and structure of the FH domains are highly conserved within the FOXO family members[34,35].

During cellular senescence induction, both FOXO4 and p53 localize to PML bodies[5]. Notably, these proteins not only co-localize, but can also physically interact, as shown by in vitro NMR spectrometry studies using purified proteins. Given the promising results obtained by us and others, targeting the FOXO4–p53 axis emerges as a promising strategy for selectively clearing of senescent cells and potentially treating age-related diseases[6–8]. However, a molecular understanding of the FOXO4–p53 interaction is crucial. Such insights will help optimize existing drugs and design drugs with enhanced potency. Beyond therapeutic applications, investigating the FOXO4–p53 interaction is pivotal for comprehending a critical mechanism that influences cell fate.

In this study, we demonstrate that the interaction between FOXO4 and p53 is mediated by specific regions: FOXO4$^{FH}$ and p53$^{TAD2}$. Additionally, the FOXO4 N-terminal disordered region and the FOXO4$^{CR3}$ show weak binding to the p53$^{DBD}$. Through extensive biophysical characterization and structure determination of the FOXO4$^{FH}$–p53$^{TAD2}$ complex, we revealed intriguing details about their interactions. Interestingly, the cell-penetrating peptide FOXO4-DRI also binds to p53$^{TAD2}$, competing with the binding of FOXO4$^{FH}$ and p53$^{DBD}$. Both p53$^{TAD2}$ and FOXO4-DRI are intrinsically disordered in solution, but upon binding, they fold synergistically. Lastly, we highlight that phosphorylation of p53$^{TAD2}$ at Ser46 and Thr55 enhances its binding affinity to FOXO4$^{FH}$ and FOXO4-DRI. Our results provide crucial molecular insights into the p53-FOXO4 interaction and shed light on the mode-of-action of FOXO4-DRI. Understanding these mechanisms is fundamental for deciphering cellular senescence regulation and developing potent inhibitors to combat age-related diseases.

## Results

### Multiple regions within p53 mediate interactions with FOXO4

To establish the specific binding sites on p53 and FOXO4 that mediate their interaction, we conducted binding studies using individual fragments of p53 and FOXO4. We started by examining the known binding of FOXO4$^{FH}$ to a p53 fragment spanning residues 1 to 312 (hereafter termed p53$^{1-312}$), which includes the TAD, the PRD and the DBD (Fig. 1a). We first identified the binding site of FOXO4$^{FH}$ in p53$^{1-312}$. Upon addition of unlabeled recombinant FOXO4$^{FH}$ to $^{15}$N-labeled p53$^{1-312}$ we observed progressive chemical shift perturbations (CSPs) of several p53$^{1-312}$ $^1$H,$^{15}$N cross-peaks, which indicated direct interaction between p53$^{1-312}$ and FOXO4$^{FH}$.

Interestingly, most of the affected peaks are localized in regions of the NMR spectrum characteristic of cross-peaks corresponding to disordered regions, suggesting that the TAD and/or PRP, rather than the DBD, mediate FOXO4 binding (Fig. 1b). To follow up on this finding, we investigated whether the N-terminal p53 region from amino acids 1 to 94 (hereafter termed p53$^{1–94}$) or the p53$^{DBD}$ alone could bind to FOXO4. The spectrum of $^{15}$N-labeled p53$^{DBD}$ remained unchanged upon addition of unlabeled FOXO4$^{FH}$, indicating absence of interactions (Supplementary Fig. 1a). However, when we added unlabeled FOXO4$^{FH}$ to a solution of $^{15}$N-labeled p53$^{1–94}$, we observed characteristic CSPs of several cross-peaks. The affected residues clustered in the region from amino acids 45 to 57, corresponding to the TAD2 (Fig. 1c). Minor CSPs were seen in the region corresponding to TAD1. To further explore whether TAD1 or TAD2 alone could independently bind to FOXO4$^{FH}$, we monitored CSPs for $^{15}$N-labeled p53$^{TAD1}$ and p53$^{TAD2}$, respectively upon addition of unlabeled FOXO4$^{FH}$. Notably, we found that p53$^{TAD2}$,

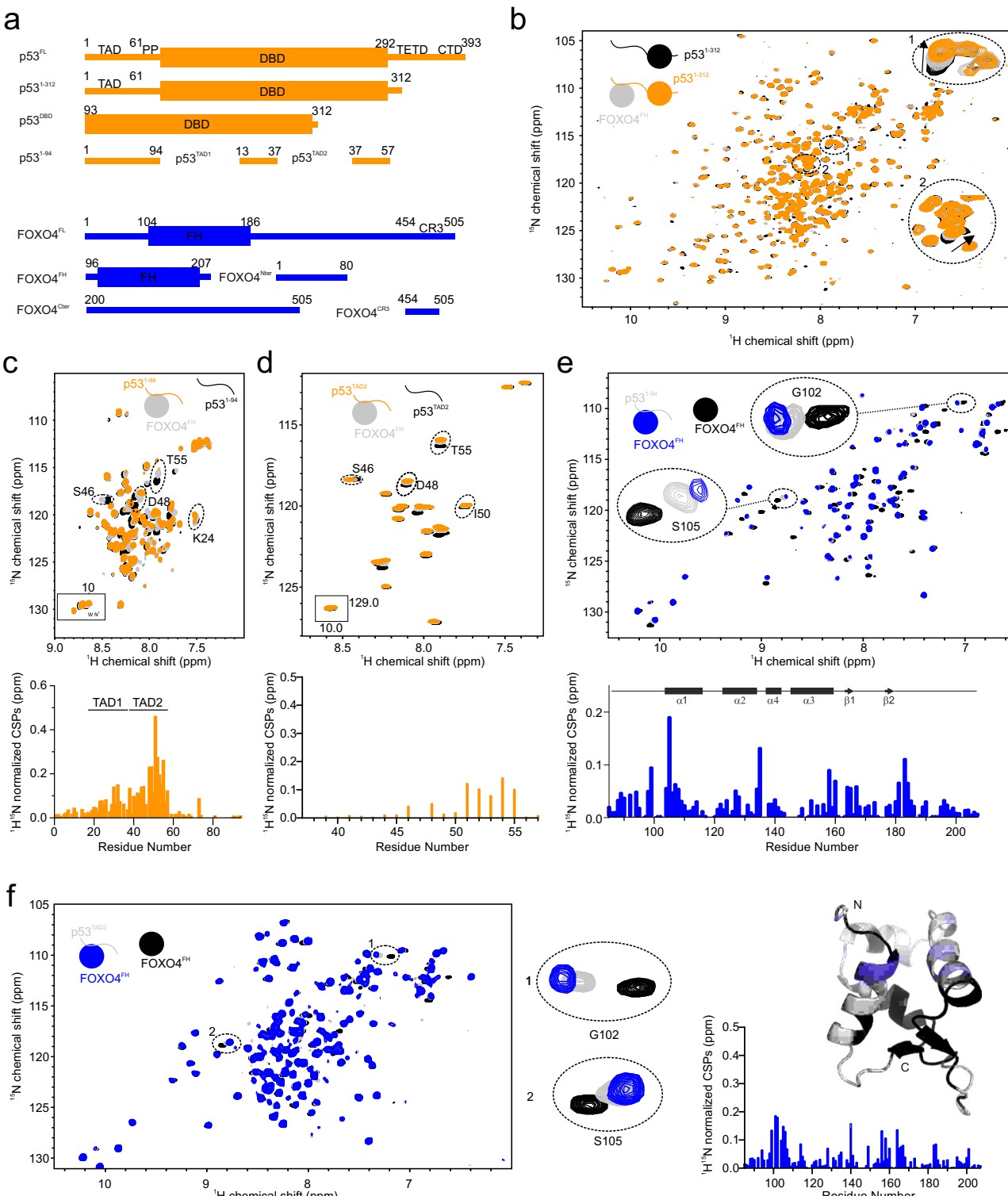

**Fig. 1 | p53^TAD2 is the primary binding site of FOXO4^FH. a** Domain architecture of p53 and FOXO4. **b** ¹H,¹⁵N HSQC spectrum of 100 µM ¹⁵N-labeled p53^1-312 in absence (black) and in presence of one stoichiometric (gray) or three stoichiometric (orange) equivalents of unlabeled FOXO4^FH. **c** ¹H,¹⁵N HSQC spectrum of 100 µM ¹⁵N-labeled p53^1-94 in absence (black) and in presence of one stoichiometric (gray) or two (orange) stoichiometric equivalents of unlabeled FOXO4^FH, with corresponding CSPs. **d** ¹H,¹⁵N HSQC spectrum of 100 µM ¹⁵N-labeled p53^TAD2 in absence (black) and in presence of one stoichiometric (gray) or two (orange) stoichiometric equivalents of unlabeled FOXO4^FH, with corresponding CSPs. **e** ¹H,¹⁵N HSQC spectrum of 100 µM ¹⁵N-labeled FOXO4^FH in absence (black) or in presence of one (gray) or two (blue) stoichiometric equivalents of unlabeled p53^1-94, with corresponding CSPs. **f** ¹H,¹⁵N HSQC spectrum of 100 µM ¹⁵N-labeled FOXO4^FH in absence (black) and in presence of one (gray) or two (blue) stoichiometric equivalents of unlabeled p53^TAD2, with corresponding CSPs.

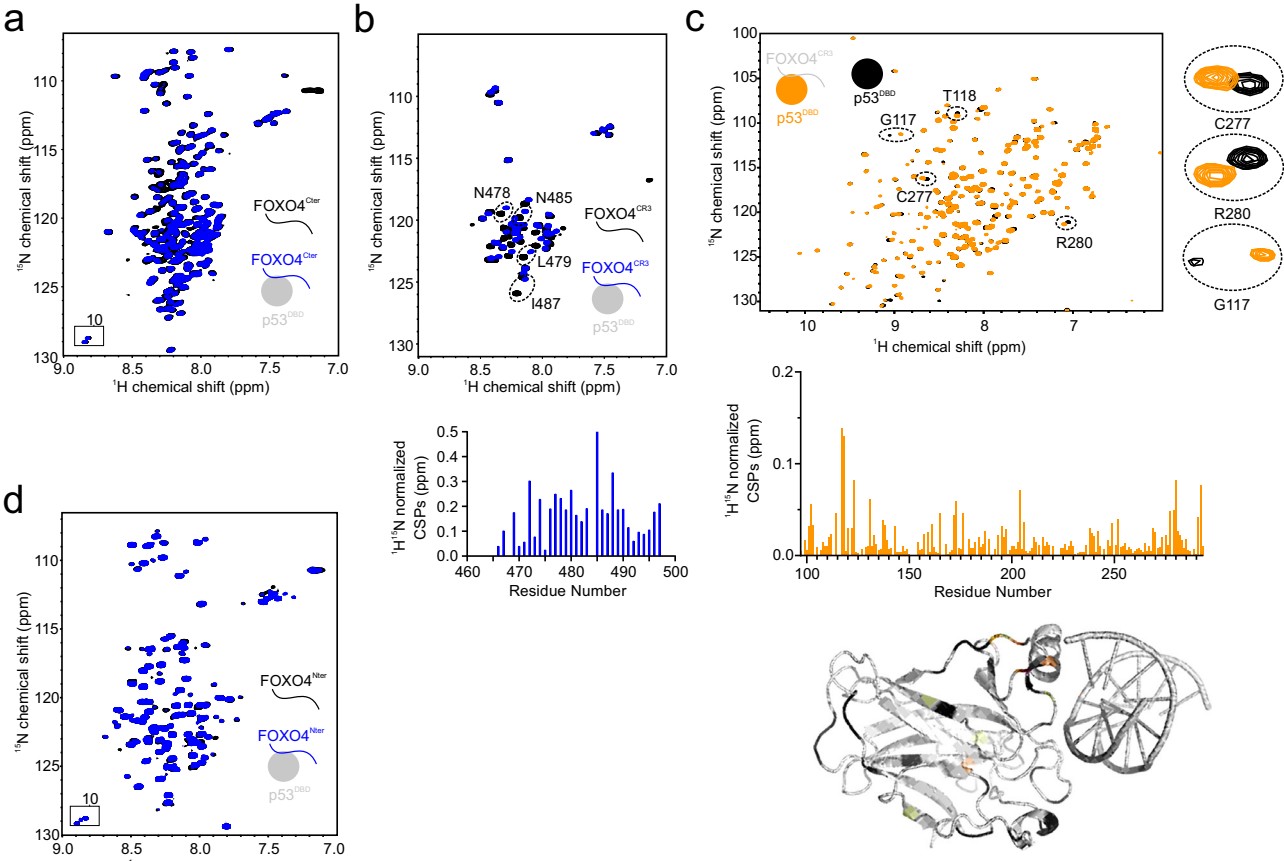

**Fig. 2 | Disordered regions of FOXO4 weakly bind to p53^DBD.** ¹H,¹⁵N HSQC spectrum of 100 μM ¹⁵N-labeled FOXO4^Cter (**a**)/FOXO4^CR3 (**b**) in the absence (black) or presence of one stoichiometric equivalent of unlabeled p53^DBD (blue). A CSP quantification is shown for FOXO4^CR3. **c** ¹H,¹⁵N HSQC spectrum of 100 μM ¹⁵N-labeled p53^DBD in the absence (black) or presence (orange) of one stoichiometric equivalent of unlabeled FOXO4^CR3 with corresponding CSP quantification. The CSPs are plotted on the structure of the p53^DBD (PDB ID 4HJE), with a color code from white to orange representing the increasing strength of the CSP. Unassigned residues are colored black. **d** ¹H,¹⁵N HSQC spectrum of 100 μM ¹⁵N-labeled FOXO4^Nter in the absence (black) and in presence of one stoichiometric equivalent of unlabeled p53^DBD (blue).

but not p53^TAD1 exhibited binding to FOXO4^FH on its own (Fig. 1d and Supplementary Fig. 1b).

We then aimed to investigate which surface of FOXO4^FH interacts with p53. To achieve this, we conducted ¹⁵N chemical shift titrations using ¹⁵N-labelled FOXO4^FH and the corresponding interacting fragments of p53. In line with our data obtained for p53^1-312, several ¹H,¹⁵N cross-peaks of ¹⁵N FOXO4^FH shifted progressively upon addition of p53^1-94 (Fig. 1e). Analyzing the CSPs, we found that residues in the disordered region preceding α-helix 1 (H1) of FOXO4^FH (Arg92, Trp101, Gly102, Ser105, and Tyr106) and regions within α-helix 3 (H3; Tyr137, Trp150, Ser153 were most affected by p53^1-94 binding (Fig. 1e). Next, we explored whether individual p53^TAD1 or p53^TAD2 alone are sufficient to induce CSPs in the spectrum of ¹⁵N-labeled FOXO4^FH and performed titrations using p53 fragments corresponding to the sequence of TAD1 and TAD2, respectively. As expected, the addition of p53^TAD2 led to CSPs of ¹H,¹⁵N cross-peaks of FOXO4^FH (Fig. 1f), whereas addition of p53^TAD1 did not cause any changes (Supplementary Fig. 1c), confirming that TAD2, but not TAD1, alone is sufficient for binding to FOXO4^FH. In line with this, we could pull-down higher amounts of endogenous p53 versus p53^ΔTAD2 in RPE-1 cells using recombinant biotinylated FOXO4^FH (Supplementary Fig. 1d).

Previously, Wang *et al.* reported that the conserved region 3 (CR3) of FOXO3 interacts with p53 by binding to the DBD[36]. Given the high conservation of the CR3 across FOXO proteins[34,35], we set out to explore whether a similar binding event occurs involving the disordered regions of FOXO4. To investigate this, we conducted NMR titrations of ¹⁵N-labeled FOXO4^Cter, which contains the CR3, with unlabeled p53 fragments. As expected, we observed the disappearance and line broadening of specific ¹H,¹⁵N FOXO4^Cter cross-peaks in the presence of p53^DBD, confirming their binding (Fig. 2a). In addition, we carried out several control experiments, which showed, as conclusion, no binding between p53^1-94 and FOXO4^Cter, p53^1-94 and FOXO4^Nter, respectively (Supplementary Fig. 2a-b). p53^DBD showed weak binding to FOXO4^Nter (Fig. 2d). Next, we explored whether the FOXO4^CR3 alone could bind to p53^DBD. By titrating ¹⁵N-labeled FOXO4^CR3 with unlabeled p53^DBD, we observed progressive CSPs in FOXO4^CR3 ¹H,¹⁵N cross-peaks (Fig. 2b). The most affected residues were localized in the region between FOXO4 amino acids 473 and 491 (Fig. 2b). Finally, we aimed to pinpoint the specific binding site of FOXO4^CR3 on the p53^DBD. Introducing unlabeled FOXO4^CR3 to a solution of ¹⁵N-labeled p53^DBD resulted in progressive CSPs of p53^DBD ¹H,¹⁵N cross-peaks (Fig. 2c). Notably, most of the affected peaks were concentrated in helix 2 of the p53^DBD, which is responsible for p53 binding to DNA (Fig. 2c)[37]. More precisely the interface on p53^DBD involves the surface, which is responsible for DNA binding. Accordingly, addition of DNA to a sample of ¹⁵N-labeled FOXO4^CR3 in complex with unlabeled p53^DBD outcompeted FOXO4^CR3 from p53^DBD (Supplementary Fig. 2c). Taken together the interaction between p53 and FOXO4 involves multiple binding events.

Our investigation into the interaction sites between p53 and FOXO4 revealed that the FOXO4^FH and the p53 TAD (transactivation domain) play key roles and that the N- and C-terminal disordered regions of FOXO4 additionally mediate weak binding to p53^DBD. Within

the p53 TAD, both subdomains exhibit CSPs, but notably, only p53$^{TAD2}$ is sufficient for binding to FOXO4$^{FH}$. The binding surface on FOXO4$^{FH}$ involves the disordered region preceding H1 and specific residues within H3. These insights deepen our understanding of the molecular details of their interaction and provide a foundation for further structural studies (*see below*).

## p53$^{TAD2}$ dynamics are affected by FOXO4$^{FH}$ binding

The p53 transactivation domain is known to be highly dynamic[38–40] and FOXO4$^{FH}$ binding could influence p53$^{TAD2}$ upon binding. Therefore, our next investigations focused on understanding how FOXO4$^{FH}$ binding affects the secondary structure and dynamics of p53. We utilized several NMR observables to gain insights: (i) secondary chemical shifts providing information about the propensity of adopting secondary structure elements, (ii) the steady-state heteronuclear $^{15}$N{$^1$H} Nuclear Overhauser Effect (NOEs) reporting on the motion of individual N-H bond vectors and providing information about the rigidity of the protein backbone, (iii) $R_1$ and $R_2$ relaxation rates reporting on overall tumbling and internal motions. We first calculated the secondary chemical shift based on $^{13}$C chemical shifts of Cα and Cβ atoms in p53$^{1–94}$, both in the presence and absence of unlabeled FOXO4$^{FH}$. Notably, even in the absence of FOXO4$^{FH}$, we observed an α-helical population of TAD1, which slightly increased in presence of FOXO4$^{FH}$. However, TAD2 shows only a mild preference for an α-helical conformation, which appears to be slightly stabilized in presence of FOXO4$^{FH}$ (Supplementary Fig. 3a). More strikingly, $^{15}$N{$^1$H} NOEs of p53$^{1–94}$ in the region from 48 to 57 increased substantially in the presence of FOXO4$^{FH}$, indicating increased rigidity. Meanwhile, residues corresponding to p53$^{TAD1}$ already showed positive $^{15}$N{$^1$H} NOEs even without binding partner, and these remained largely unchanged in the presence of FOXO4$^{FH}$ (Supplementary Fig. 3b). $R_1$ relaxation rates observed in free p53$^{1–94}$ and in complex with FOXO4$^{FH}$ are highly similar indicating that the sub-ns motions and rotational tumbling of p53$^{1–94}$ are only slightly affected by FOXO4$^{FH}$ binding (Supplementary Fig. 3c). In contrast, $R_2$ relaxation rates of the TAD1 and TAD2 increased substantially in the presence of FOXO4$^{FH}$ (Supplementary Fig. 3d). This is in line with the fact that the processes driving $R_2$ relaxation arise from interactions between nuclear spins within the molecule and are sensitive to slower motions, such as internal motions (e.g., side-chain dynamics, backbone flexibility) and conformational changes occurring upon FOXO4$^{FH}$ binding.

Taken together, our observations provide valuable insights into binding of the FOXO4$^{FH}$ to the TAD of p53. The binding event enhances the rigidity of both, TAD1 and TAD2, and increases the α-helical propensity of those regions.

## The FOXO4$^{FH}$·p53$^{TAD2}$ complex structural model reveals overlapping binding surfaces with DNA

To develop potent inhibitors for protein-protein interactions, such as FOXO4-p53, understanding the three-dimensional structure of the protein complex is crucial. Our focus centered on the TAD2 region of p53, which is both necessary and sufficient for binding to FOXO4$^{FH}$. Unfortunately, attempts to crystallize the complex were unsuccessful. Instead, we generated a structural model of FOXO4$^{FH}$ in complex with p53$^{TAD2}$ using NMR-based structural restraints and the available structure of the FOXO4$^{FH}$ as inputs. However, due to intermediate exchange causing extensive line broadening of residues located within the binding interface, we obtained only a few nuclear Overhauser effect (NOE)-based distance restraints. To supplement this, we introduced 7 covalent paramagnetic labels at various positions within FOXO4$^{FH}$ and p53$^{TAD2}$ and obtained paramagnetic relaxation enhancements (PREs) and PRE-based distance restraints for the structure calculations (Supplementary Fig. 4a–c). Figure 3a shows an overlay of the top 10 lowest-energy models from our ensemble.

Furthermore, to refine the experimentally determined structural model and explore interaction dynamics, we conducted extensive molecular dynamics (MD) simulations. These simulations utilized the lowest-energy solution of the NMR-derived ensemble of the p53$^{TAD2}$-FOXO4$^{FH}$ complexes as our starting geometry (see methods). Within the complex, p53$^{TAD2}$ residues 50 to 54 displayed increased rigidity upon binding and consistently maintain an α-helical conformation throughout the MD calculations (Fig. 3b). Interestingly, the helical content for Trp53/Phe54 increased, while the region preceding the p53$^{TAD2}$ helix remained extended (as indicated by NMR-derived secondary chemical shifts and MD Ramachandran data; Fig. 3b). Furthermore, our NMR relaxation data corroborate that p53$^{TAD2}$ retains flexibility upon binding to FOXO4$^{FH}$. This flexibility is reflected in the high root mean square deviation (RMSD) and root mean square fluctuation (RMSF) of p53 residues 46–55 during the MD simulations (Fig. 3c). In contrast, the well-folded portion of FOXO4$^{FH}$ exhibits low RMSD/RMSF.

To better understand the dynamics of p53$^{TAD2}$ when bound to FOXO4$^{FH}$, we used Markov state models to analyze the conformational space of p53$^{TAD2}$ both in isolation and in complex with FOXO4$^{FH}$. As anticipated, several minima were identified, revealing that the binding of p53$^{TAD2}$ to FOXO4$^{FH}$ alters the distribution of microstates within the two time-lagged independent components (ICs; Supplementary Fig. 5a–c). Notably, binding to FOXO4$^{FH}$ reduces dispersion between IC1 and IC2, resulting in more restricted conformational changes. This suggests that p53$^{TAD2}$ becomes increasingly rigid upon interaction with FOXO4$^{FH}$.

Figure 3d shows the structural model of the major conformational cluster obtained from MD simulations. In this complex, p53$^{TAD2}$ extensively interacts with FOXO4$^{FH}$ via a network of hydrophobic and charge-based interactions (Fig. 3d). In line with the CSP data (Fig. 1e) and supported by the PRE data, the recognition of p53$^{TAD2}$ by FOXO4$^{FH}$ involves the disordered N/C-terminal regions before/after the Forkhead Domain, as well as α-helix 3. Several regions within p53$^{TAD2}$ appear important for FOXO4$^{FH}$ binding (Fig. 3d): i) the negatively charged p53 region preceding the helical part (Asp41, Asp42, Asp47, Asp48) is in proximity of the positively charged FOXO4 Arg155 and His156 with H3, ii) p53 hydrophobic residues Ile50, Trp53 and Phe54 form a cluster and contact FOXO4 Ala101, and iii) the negatively charged p53 Glu51 and Glu56 and Asp57 region following the helical part are in close proximity with the positively region preceding the FOXO4$^{FH}$ (Arg98, Arg99). Overall, FOXO4$^{FH}$ provides a positively charged surface and hydrophobic anchors for the negatively charged/hydrophobic p53$^{TAD2}$ (Fig. 3e). In agreement, an increase of the salt concentration or alanine mutations of hydrophobic residues in TAD2 impair binding of p53$^{1–94}$ to FOXO4$^{FH}$ (Supplementary Fig. 6a–g, Table 1).

Comparison of our structural models with the 3D structure of the FOXO4 FH domain bound to DNA (PDB: 3L2C [41]) shows that the binding surfaces of p53$^{TAD2}$ and DNA on the FH domain overlap, suggesting mutually exclusive interactions (Fig. 3f). Furthermore, we compared the FOXO4$^{FH}$–p53$^{TAD2}$ complex structural model with the structure of the FOXO4$^{FH}$–FOXO4$^{CR3}$ complex structure determined by us recently[42]. Strikingly, both FOXO4$^{CR3}$, which is part of its transactivation domain, and p53$^{TAD2}$ bind to similar regions within FOXO4$^{FH}$ (Fig. 3f).

Summarizing, the interaction between the TAD2 of p53 and FOXO4 FH domain triggers the formation of a transient alpha helix in TAD2 and is mediated by electrostatic and hydrophobic contacts. The interface on FOXO4$^{FH}$ covered by p53$^{TAD2}$ overlaps with the one bound to DNA indicating a competition between the binding events. Accordingly, addition of a FOXO4 target DNA sequence (CTDSP2) can efficiently compete out FOXO4$^{FH}$ from p53$^{1–94}$ (Supplementary Fig. 2d).

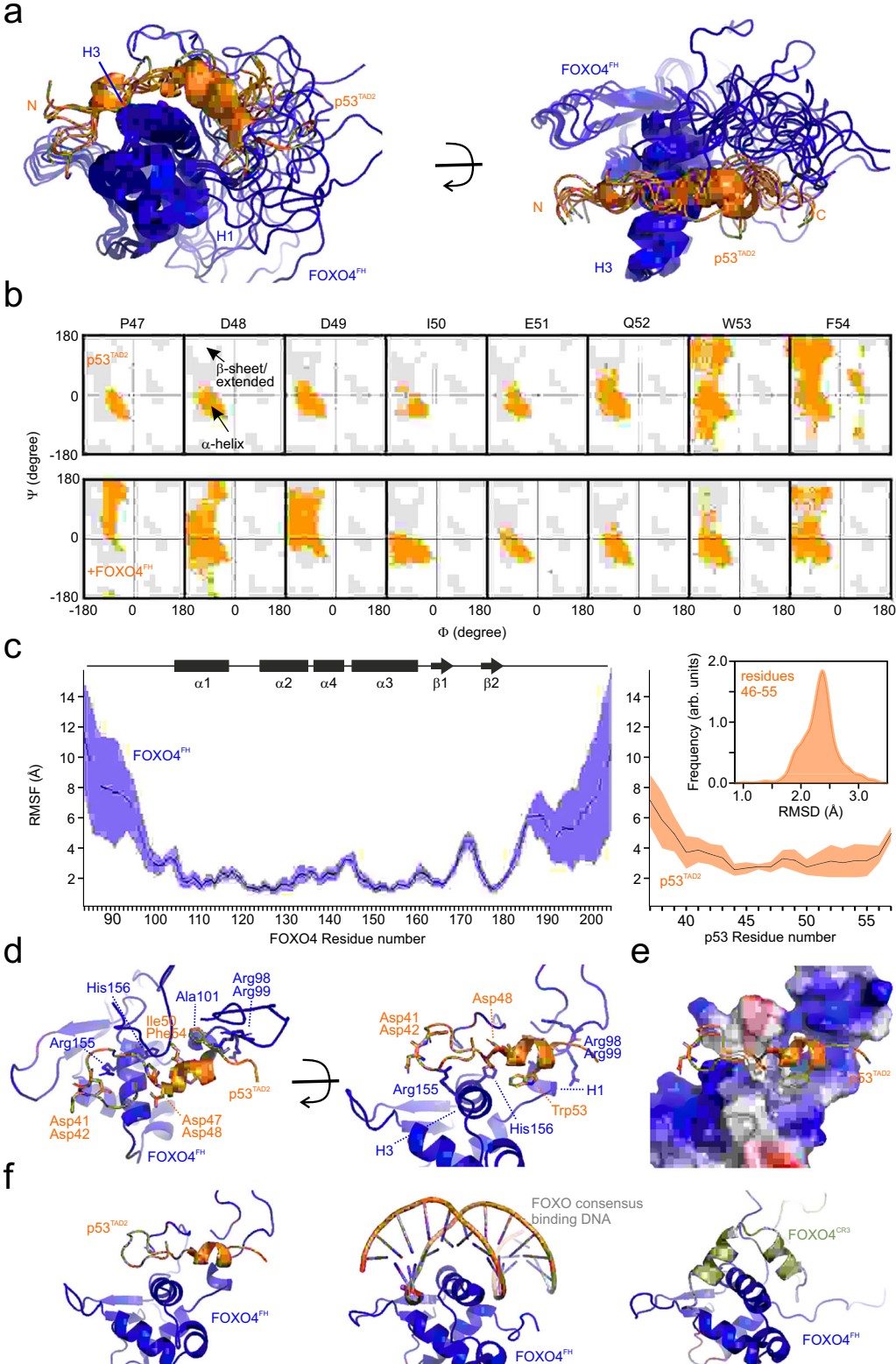

**Fig. 3 | Structural analysis of the FOXO4^FH – p53^TAD2 complex. a** Ensemble of ten lowest energy structural models of p53^TAD2 (orange) bound to FOXO4^FH (dark blue). Secondary structure elements and termini are indicated. **b** Ramachandran plot for residues 47–54 in p53^TAD2 along the different MD simulations of p53^TAD2 alone and in complex with FOXO4^FH, respectively. **c** Mean values of RMSD and RMSF of FOXO4^FH and p53^TAD2, respectively, in the FOXO4^FH-p53^TAD2 complex along the different MD simulations. The SD is shown as a shadow. **d** Structural details of the FOXO4^FH-p53^TAD2 interaction. **e** Complementary surface charge distribution of FOXO4^FH and key p53^TAD2 residues involved in the interaction (f) Comparison of the FOXO4^FH-p53^TAD2 with the crystal structure of FOXO4^FH (dark blue) bound to DNA (PDB: 3L2C) and the structure of the FOXO4^FH-FOXO4^CR3 complex[42].

**Table 1 | Thermodynamic parameters of ITC titrations**

| Cell | Syringe | N | $K_D$ (µM) | $\Delta H$ (kcal mol⁻¹) | $-T\Delta S$ (kcal mol⁻¹) |
|---|---|---|---|---|---|
| p53$^{1-94}$ b | FOXO4$^{FH}$ | 0.79 ± 0.17 | 2.5 ± 0.6 | 6.3 ± 0.3 | −13.9 ± 0.5 |
| p53$^{1-94}$ I50A b | FOXO4$^{FH}$ | 0.83 ± 0.15 | 6.3 ± 0.8 | 4.6 ± 0.7 | −11.9 ± 0.5 |
| p53$^{1-94}$ W53A b | FOXO4$^{FH}$ | 0.87 ± 0.10 | 8.2 ± 1.8 | 5.6 ± 0.4 | −12.6 ± 0.2 |
| p53$^{1-94}$ F54A b | FOXO4$^{FH}$ | 0.83 ± 0.05 | 5.1 ± 2.4 | 5.7 ± 0.5 | −12.9 ± 1.0 |
| p53$^{1-94}$ IFWtoA b | FOXO4$^{FH}$ | 0.95 ± 0.11 | 16.7 ± 9.2 | 6.4 ± 2.1 | −13.0 ± 1.9 |
| p53$^{1-94}$ high salt | FOXO4$^{FH}$ | n.d. | n.d. | n.d. | n.d. |
| p53$^{1-94}$ a | FOXO4-DRI | 0.90 | 0.4 ± 0.3 | 5.9 ± 0.1 | −14.6 |
| p53$^{TAD2}$ a | FOXO4$^{FH}$ | n.d. | n.d. | n.d. | n.d. |
| p53$^{TAD2pS46}$ a | FOXO4$^{FH}$ | 1.48 | 30.6 ± 4.8 | −1.6 ± 0.1 | −5.0 |
| p53$^{TAD2pT55}$ a | FOXO4$^{FH}$ | 0.95 | 4.6 ± 0.2 | −3.1 ± 0.1 | −4.2 |
| p53$^{TAD2pS46T55}$ a | FOXO4$^{FH}$ | 1.09 | 5.7 ± 0.3 | −4.4 ± 0.1 | −2.7 |
| p53$^{TAD2}$ a | FOXO4-DRI | 0.90 | 18.5 ± 3.6 | 0.2 ± 0.1 | −6.7 |
| p53$^{TAD2pS46}$ a | FOXO4-DRI | 0.71 | 10.5 ± 1.5 | −1.8 ± 0.1 | −5.0 |
| p53$^{TAD2pT55}$ a | FOXO4-DRI | 0.74 | 5.8 ± 1.2 | −4.7 ± 0.1 | −5.0 |
| p53$^{TAD2pS46T55}$ a | FOXO4-DRI | 1.11 | 4.9 ± 0.3 | −4.5 ± 0.1 | −2.6 |

ªThe reported errors correspond to the SD of the fit.
ᵇThe reported errors correspond to the SD to the mean values of the triplicate experiments ($n = 3$).

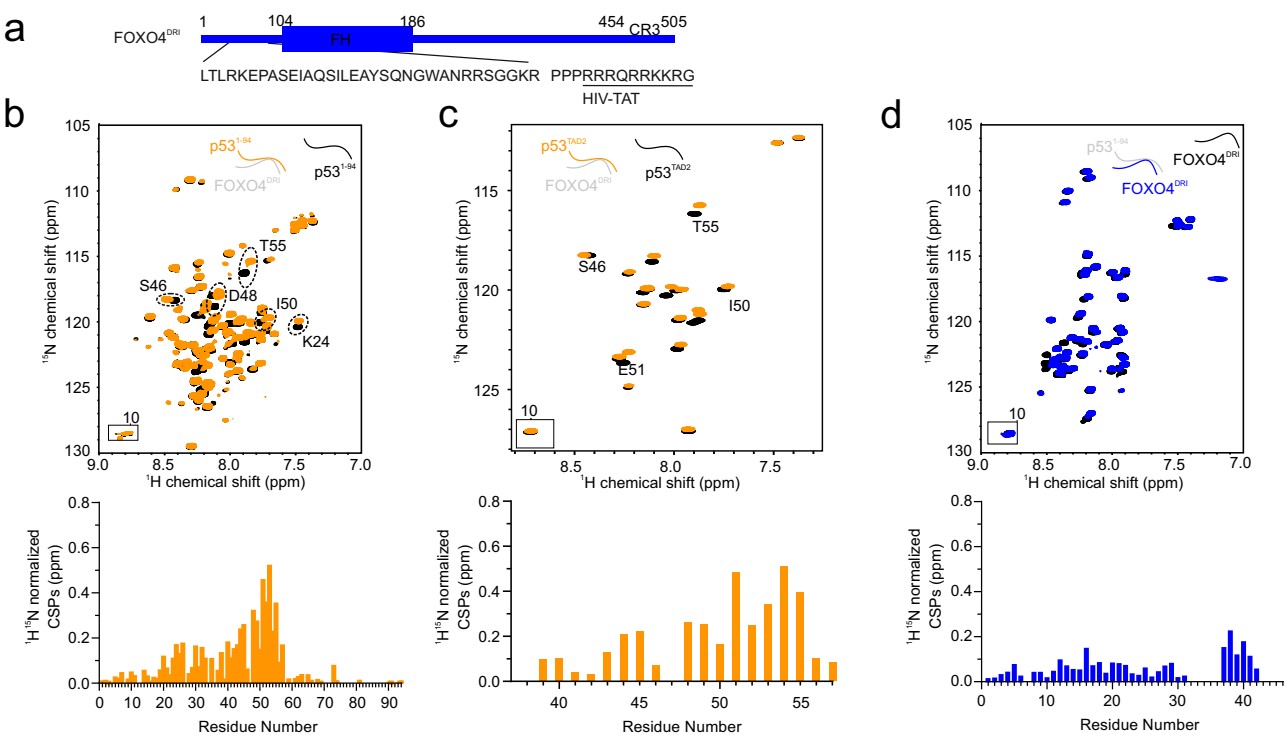

**Fig. 4 | FOXO4-DRI directly binds to p53$^{TAD}$. a** Sequence of FOXO4-DRI. **b** ¹H,¹⁵N HSQC spectra of ¹⁵N-labeled p53$^{1-94}$ in the absence (black) or presence of one stoichiometric equivalent of FOXO4-DRI (orange) and corresponding CSPs. **c** ¹H,¹⁵N HSQC spectra of ¹⁵N-labeled p53$^{TAD2}$ in the absence (black) or presence of one stoichiometric equivalent of FOXO4-DRI (orange). **d** ¹H,¹⁵N HSQC spectra of ¹⁵N-labeled FOXO4-LRI in the absence (black) or presence of one stoichiometric equivalent of $D$-p53$^{TAD2}$ (blue) and corresponding CSPs.

## FOXO4$^{DRI}$ competes with FOXO4$^{FH}$ for p53$^{TAD2}$ binding

In 2018 we discovered a remarkable *D*-retro-inverso cell-penetrating peptide derived from the FOXO4$^{FH}$ domain[5]. This peptide efficiently competes with the p53–FOXO4 interaction and triggers apoptosis in senescent cells. The peptide's sequence originates from the N-terminal disordered region and α-helix 1 of the FOXO4$^{FH}$ domain and was fused to the HIV-TAT to enhance cell permeability (Fig. 4a).

To pinpoint the binding site of FOXO4-DRI on p53, we conducted a series of NMR titrations using different fragments of p53. The spectrum of recombinant ¹⁵N-labeled p53$^{1-94}$ exhibited CSPs in presence of FOXO4-DRI, confirming binding. Interestingly, FOXO4-DRI binds the same residues as FOXO4$^{FH}$, consistent with its competition against the native p53-FOXO4 interaction observed in cells and in vivo (Fig. 4b)[5]. Notably, the CSPs are even more pronounced for FOXO4-DRI compared to FOXO4$^{FH}$, reflecting its potent interference with the native interaction (Fig. 4b). In agreement, addition of FOXO4-DRI can efficiently compete out p53$^{TAD2}$ from FOXO4$^{FH}$, as indicated by the chemical shift of ¹H,¹⁵N FOXO4$^{FH}$ cross-peaks that return from a p53

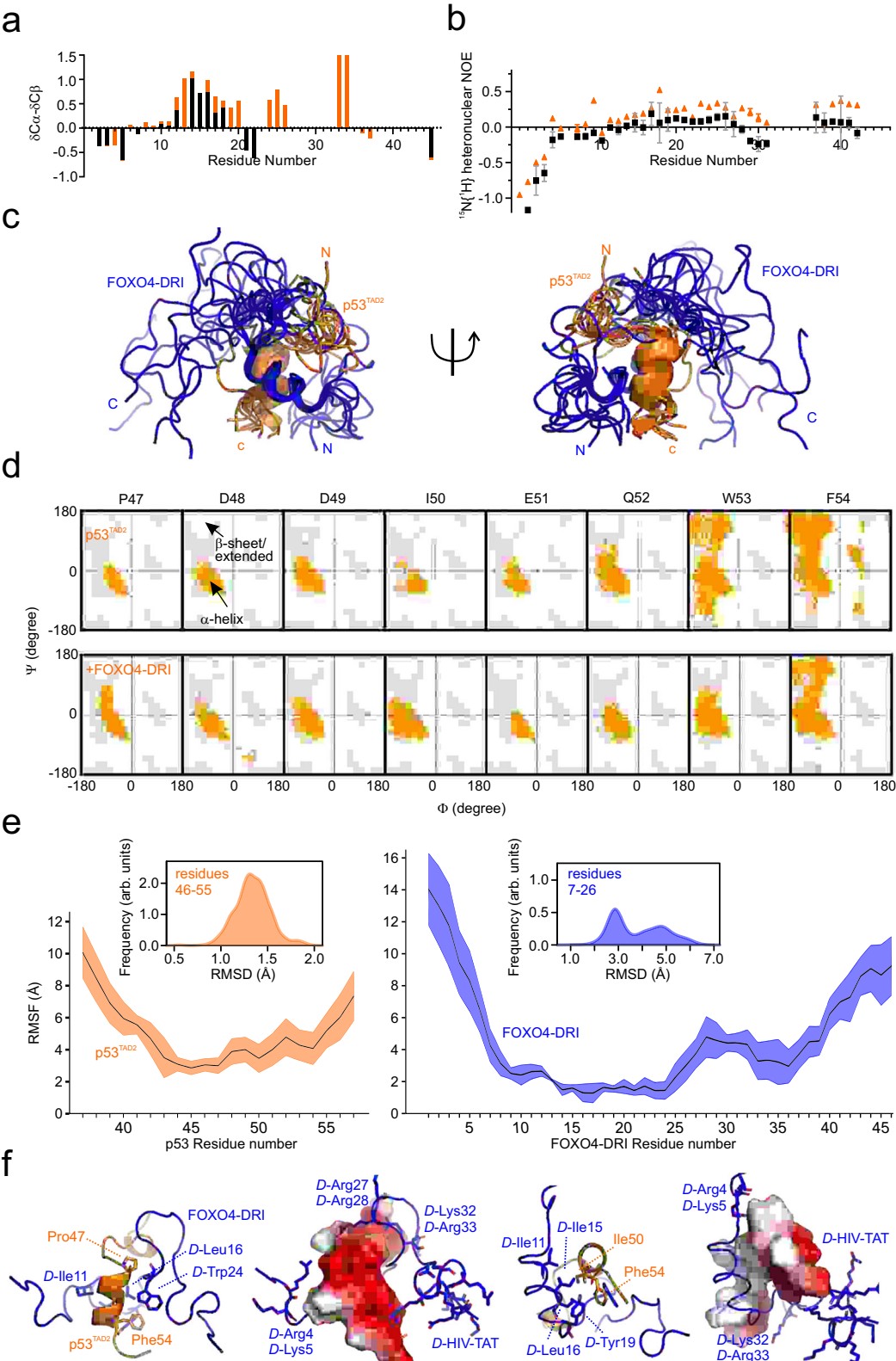

**Fig. 5 | FOXO4-DRI and p53<sup>TAD2</sup> synergistically fold upon binding. a** Plot of difference between the secondary $^{13}C$ chemical shifts of Cα and Cβ nuclei of FOXO4-LRI in the absence (black) and presence of 400 µM unlabeled $D$-p53$^{TAD2}$ (orange). **b** $^{15}N\{^1H\}$-NOE of 300 µM FOXO4-LRI in the absence (black) and in presence of 400 µM unlabeled $D$-p53$^{TAD}$ (orange). Error bars were calculated based on the standard deviation of noise in the saturated and unsaturated spectra and using error propagation. **c** Ensemble of ten lowest energy structural models of p53$^{TAD2}$

(orange) bound to FOXO4-DRI (blue). **d** Ramachandran plot for residues 47–54 in p53$^{TAD2}$ along the different MD simulations of p53$^{TAD2}$ alone and in complex with FOXO4-DRI, respectively. **e** Mean values of RMSD and RMSF of p53$^{TAD2}$ and FOXO4-DRI, respectively, in the FOXO4-DRI-p53$^{TAD2}$ complex along the different MD simulations. The SD is shown as a shadow. **f** Structural details of the FOXO4-DRI-p53$^{TAD2}$ interaction. Complementary surface charge distributions of p53$^{TAD2}$ and key FOXO4-DRI residues involved in the interaction.

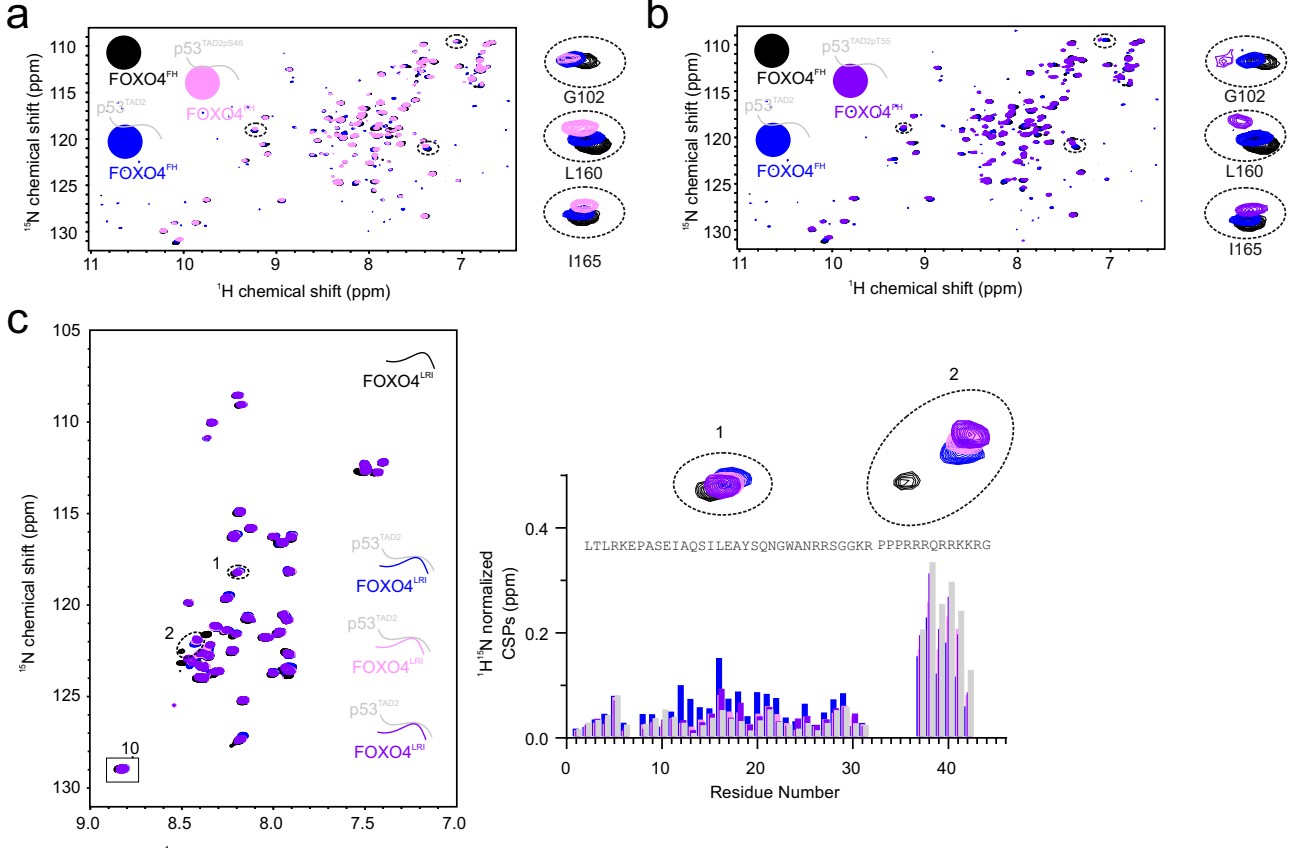

**Fig. 6 | Phosphorylation of p53^TAD2 enhances its affinity for FOXO4^FH and FOXO4-DRI. a** Overlay of ^1H,^15N HSQC spectra of ^15N-labeled FOXO4^FH in the absence (black) or presence of one stoichiometric equivalent of p53^TAD2 (blue) or p53^TAD2pS46 (pink). **b** Overlay of ^1H,^15N HSQC spectra of ^15N-labeled FOXO4^FH in the absence (black) or presence of one stoichiometric equivalent of p53^TAD2 (blue) or p53^TAD2pT55 (violet). **c** Overlay of ^1H,^15N HSQC spectra of ^15N-labeled FOXO4-LRI in absence (black) or presence of two stoichiometric equivalent of p53^TAD2 (bleu), p53^TAD2pS46 (pink) or p53^TAD2pT55 (violet) and associated CSPs.

bound state to a p53 free state upon addition of FOXO4-DRI (Supplementary Fig. 7a). Notably, no CSPs were observed for p53^DBD upon addition of FOXO4-DRI, confirming the lack of binding (Supplementary Fig. 7b). We further validated binding of FOXO4-DRI to p53^1–94 by using isothermal titration calorimetry (ITC). In line with the NMR results, the affinity of FOXO4-DRI for p53^1–94, with an associated $K_d$ of 400 ± 280 nM, is 5-fold stronger than the affinity of FOXO4^FH for p53^1–94 (Supplementary Fig. 6g, Table 1). To disentangle the contributions of p53^TAD1 and p53^TAD2 we performed a series of NMR titrations using ^15N-labeled p53^TAD2 and added increasing amounts of FOXO4-DRI. Observable CSPs were induced in the ^1H,^15N cross-peaks of ^15N-labeled p53^TAD2 (Fig. 4c). Summarizing, this demonstrates that, like FOXO4^FH, the primary binding site for FOXO4-DRI lies within the p53^TAD2. In this line, we could pull-down higher amount of endogenous p53 versus p53^ΔTAD2 in RPE-1 cells using recombinant biotinylated FOXO4^DRI (Supplementary Fig. 7c).

To gain deeper insights into the p53^TAD2-FOXO4-DRI interaction from the side of FOXO4-DRI, we expressed and purified the *L*-amino acid version of the FOXO4-DRI sequence in *E. coli*, which we will refer to as FOXO4-LRI. Since FOXO4-LRI consists of *L*-amino acids (as opposed to *D*-amino acids), we used synthetic p53 peptides composed of *D*-amino acids corresponding to the p53^TAD2 sequence. This approach preserved proper stereochemistry in a mirror image fashion. Upon titrating *D*-p53^TAD2 to recombinant ^15N-labeled FOXO4-LRI, we observed CSPs in FOXO4-LRI ^1H,^15N cross-peaks, confirming direct interaction (Fig. 4d). Quantification of the CSPs revealed affected residues within the range of amino acids 8–22, as well as residues within the HIV TAT (Fig. 4d).

Taken together our data demonstrates, that FOXO4-DRI binds to the same region as FOXO4^FH with a higher affinity, thereby disrupting the interaction between FOXO4^FH and p53^TAD.

## FOXO4^DRI and p53^TAD2 synergistically fold upon binding

Building upon our observations made for FOXO4^FH binding to p53^1–94, we then aimed to explore how FOXO4-DRI binding affects the secondary structure and dynamics of p53^1–94. Our investigation involved assessing several key parameters - the secondary chemical shift, steady-state ^15N{^1H} NOEs, $R_1$, and $R_2$ relaxation rates for p53^1–94 in presence and absence of FOXO4-DRI. Upon binding FOXO4-DRI, the TAD1 and TAD2 regions exhibit a stronger tendency to adopt α-helical conformations (Supplementary Fig. 7d). This increased helical content aligns with enhanced rigidity in these regions, as evidenced by increased ^15N{^1H} NOEs of p53^1–94 in presence of FOXO4-DRI (Supplementary Fig. 7e). As also observed for FOXO4^FH binding, the $R_1$ relaxation rates of p53^1–94 remained unaltered in presence of FOXO4-DRI (Supplementary Fig. 7f), whereas $R_2$ relaxation rates in the TAD1 and TAD2 regions increased substantially (Supplementary Fig. 7g).

Turning our attention to FOXO4-LRI, we observed a similar outcome upon ligand binding. Initially, in the free state, FOXO4-LRI exhibited secondary chemical shifts characteristic of an intrinsically disordered peptide. However, upon binding to *D*-p53^TAD2, several FOXO4-LRI residues shifted towards positive secondary chemical shifts, indicative for α-helix formation (Fig. 5a). Furthermore, ^15N{^1H} NOEs revealed positive values from residues 17 to 27 already in the free state, and these values further increased upon binding to *D*-p53^TAD2. These findings suggest higher rigidity within the complex (Fig. 5b).

To determine the solution structural model of the p53$^{TAD2}$-FOXO4-DRI complex using NMR spectroscopy, we recorded several sets of NMR experiments on two types of complexes: $^{13}C,^{15}N$-labelled p53$^{TAD2}$-FOXO4-DRI and $^{13}C,^{15}N$-labelled FOXO4-LRI $D$-p53$^{TAD2}$ (see methods). Although the complex remains highly dynamic upon binding, we were able to obtain a set of distance restraints from these experiments. These restraints, along with secondary structure information derived from chemical shifts, were used for structure calculations. An ensemble of the ten lowest energy structural models is shown in Fig. 5c.

Furthermore, to refine the experimentally determined structural model and explore interaction dynamics, we conducted extensive MD simulations. These simulations utilized the lowest-energy solution of the NMR-derived ensemble of the p53$^{TAD2}$-FOXO4-DRI complexes as our starting geometry (see methods). Within the complex, p53$^{TAD2}$ residues 47 to 54 displayed increased rigidity upon binding and consistently maintain an α-helical conformation throughout the MD calculations (Fig. 5d). Interestingly, and as observed for FOXO4$^{FH}$ binding, the helical content for Trp53/Phe54 increased. Furthermore, MD revealed decreased flexibility of p53$^{TAD2}$ in complex with FOXO4-DRI compared to the FOXO4$^{FH}$-bound state with RMSDs of 1.4 and 2.4 Å, respectively (Fig. 5e). In line with the NMR relaxation data, the regions outside the core interaction (formed by p53 residues 45-55 and FOXO4-DRI residues 7–26) site displayed high flexibility in MD, as reflected in the high RMSDs and RMSFs during the MD simulations. Markov state modeling of p53$^{TAD2}$ both in isolation and in complex with FOXO4-DRI displayed several minima, revealing that the binding of p53$^{TAD2}$ to FOXO4-DRI alters the distribution of microstates within the two time-lagged independent components (Supplementary Fig. 8a–c). As in case of FOXO4$^{FH}$ binding, binding to FOXO4-DRI reduces dispersion between IC1 and IC2, resulting in more restricted conformational changes and increasing rigidity in the bound state.

Figure 5f shows the structural model of the major conformational cluster obtained from MD simulations. Both p53$^{TAD2}$ and FOXO4-DRI form short helices at the binding interface. A small hydrophobic pocket is created by Pro47, Ile50 and Phe54 of p53$^{TAD2}$ as well as by Ile11, Ile15, Leu16, Tyr19 and Trp24 of FOXO4-DRI. Additionally, multiple electrostatic contacts involve positively charged arginine residues of FOXO4-DRI, although these could not be assigned unambiguously. The electrostatic surface of FOXO4-DRI is predominantly positive, attracting negatively charged residues within p53$^{TAD2}$. Interestingly, although the binding mode to p53$^{TAD2}$ is similar compared to FOXO4$^{FH}$ (driven by hydrophobic and electrostatics interactions), there are some differences in the binding mode of FOXO4-DRI when compared to FOXO4$^{FH}$. In the presence of the former ligand, p53$^{TAD2}$ Ser46 interacts intra-molecularly with Asp48, what increases the alpha-helicity of p53$^{TAD2}$ for residues 48-53.

Summarizing, we provide three-dimensional structural models of the p53$^{TAD2}$–FOXO4-DRI complex, which is of importance for optimizing cell-penetrating peptides to target p53.

### Phosphorylation of p53$^{TAD2}$ enhances FOXO4$^{FH}$ and FOXO4-DRI binding

The activity of p53 is tightly controlled by post-translational modifications[10]. Within p53's N-terminal TAD, multiple phosphorylation sites exist, including Ser15, Thr18 and Ser20 in TAD1 and Ser46 and Thr55 in TAD2. Recent studies revealed that TAD2 phosphorylation influences p53 DNA binding by enhancing the intramolecular interaction between TAD2 and the DBD[14].

Based on that, we hypothesized that the phosphorylation of p53$^{TAD2}$ also impacts its binding to FOXO4$^{FH}$ and FOXO4-DRI. To explore this, we employed NMR spectroscopy and ITC. We used recombinantly expressed $^{15}N$-labeled FOXO4$^{FH}$ and titrated it with synthetic peptides corresponding to the sequence of p53$^{TAD2}$. These peptides were either unphosphorylated or phosphorylated at Ser46 (p53$^{TAD2pS46}$) or Thr55 (p53$^{TAD2pT55}$) (Fig. 6a, b). Phosphorylation at either

of the sites led to stronger CSPs in the $^{1}H,^{15}N$ cross-peaks of $^{15}N$-labeled FOXO4$^{FH}$, indicating enhanced binding. Next, we investigated the effect of p53$^{TAD2}$ phosphorylation on FOXO4-DRI binding. Using recombinantly expressed $^{15}N$-labeled FOXO4-LRI, we titrated it with synthetic peptides composed of $D$-amino acids corresponding to p53$^{TAD2}$. These peptides were either unphosphorylated or phosphorylated at Ser46 ($D$-p53$^{TAD2pS46}$) or at Thr55 ($D$-p53$^{TAD2pT55}$) or at both sites ($D$-p53$^{TAD2pS46T55}$). Interestingly, quantification of the CSPs in the $^{1}H,^{15}N$ cross-peaks of $^{15}N$-labeled FOXO4-LRI revealed that residues corresponding to the HIV-TAT were affected more strongly by p53$^{TAD2}$ phosphorylation at any site. Most other residues showed stronger CSPs in the presence of unphosphorylated p53$^{TAD2}$ (Fig. 6c).

To complement the NMR data and to further investigate the impact of p53$^{TAD2}$ phosphorylation on its interaction with FOXO4$^{FH}$ and FOXO4-DRI, we conducted a series of ITC experiments. We titrated p53$^{TAD2}$, p53$^{TAD2pS46}$, p53$^{TADpT55}$, and p53$^{TAD2pS46T55}$ into solutions of FOXO4-DRI and FOXO4$^{FH}$, respectively. The interaction of p53$^{TAD2}$ with FOXO4-DRI as well as with FOXO4$^{FH}$ happens spontaneously endothermic. The signal-to-noise, however, was too low to reliably fit the binding curve in case of FOXO4$^{FH}$ (Supplementary Fig. 9a). We determined the binding affinity of unphosphorylated p53$^{TAD2}$ for FOXO4-DRI with a dissociation constant of $18.5 \pm 3.6\,\mu M$ (Supplementary Fig. 9e). We note that the affinity of FOXO4-DRI for p53$^{TAD2}$ is weaker than for p53$^{1–94}$, indicating a contribution of TAD1 in binding. In agreement with the NMR data, we observed an increase in ITC-derived binding affinities of phosphorylated p53$^{TAD2}$ peptides for FOXO4$^{FH}$ with associated dissociation constants of $30.6 \pm 4.8\,\mu M$, $4.6 \pm 0.2$, and $5.7 \pm 0.3$, for p53$^{TAD2pS46}$, p53$^{TAD2pT55}$, and p53$^{TAD2pS46T55}$, respectively (Supplementary Fig. 9b–d). Also, in line with our NMR data we observed an increase in ITC-derived binding affinities of phosphorylated p53$^{TAD2}$ peptides for FOXO4-DRI with associated dissociation constants of $10.5 \pm 1.5\,\mu M$, $5.8 \pm 1.2$, and $4.9 \pm 0.3$, for p53$^{TAD2pS46}$, p53$^{TADpT55}$, and p53$^{TAD2pS46T55}$ respectively (Supplementary Fig. 9f–h). Interestingly, the interaction of phosphorylated p53$^{TAD2}$ with FOXO4$^{FH}$ and FOXO4-DRI happens spontaneously exothermic. Enthalpic contributions play a larger role in these interactions compared to unphosphorylated p53$^{TAD2}$.

Prompted by the NMR and ITC data we sought to gain deeper insights into how p53 phosphorylation affects its interaction with FOXO4$^{FH}$ and FOXO4-DRI. Therefore, we simulated free p53$^{TAD2}$ and complexes between p53$^{TAD2}$ with FOXO4$^{FH}$ and p53$^{TAD2}$ with FOXO4-DRI, considering alternative- or simultaneous phosphorylation of Ser46 and Thr55 (Supplementary Figs. 10 and 11). Importantly, when both residues (pS46 and pT55) are phosphorylated, p53$^{TAD2}$ loses helicity in the absence of binding partners. In contrast, p53$^{TAD2}$ maintains a α-helical conformation when bound to either FOXO4$^{FH}$ or FOXO4-DRI. This observation may explain the one-order magnitude difference in dissociation constants ($K_D$s) observed in ITC for both ligands, in addition to the nature of the interactions. Phosphorylation events as well as binding to either ligand alter the population of Ser46 and Thr55 in the Ramachandran plot (Supplementary Fig. 12). First, the population becomes less dispersed and it is restricted to smaller regions. Second, both residues tend to populate the region associated with ß-strands. And third, pT55 adopts an α-helical conformation only upon binding to FOXO4-DRI. In contrast, pSer46 does not exhibit helical content in any case, and pThr55 remains unstructured when bound to FOXO4$^{FH}$.

Our data underscores that phosphorylation of p53$^{TAD2}$ enhances binding affinity for both FOXO4$^{FH}$ and FOXO4-DRI. Notably, phosphorylation of Thr55 substantially strengthens this affinity.

## Discussion

Here, we identified interactions between multiple regions of FOXO4 and p53. Specifically, the FOXO4$^{FH}$ (Forkhead domain) and p53$^{TAD2}$ (transactivation domain 2) emerged as the primary interaction sites. We also detected weaker interactions involving FOXO4$^{Nter}$, the

FOXO4[CR3] and the p53[DBD] (DNA-binding domain of p53). Interestingly, and in apparent contrast to the described interaction between FOXO3 and p53[36], we did not observe any direct interaction between the p53[DBD] and FOXO4[FH], despite the large similarity of the FH domain within the FOXO family[34,35]. However, our findings do align with a study demonstrating interactions between FOXO3[CR3] and the p53[DBD][23,24], which is consistent with our observations for FOXO4. Considering the remarkable sequence conservation within the FOXO family's FH, it is plausible that similar interactions with p53 may exist in other FOXO family members.

Emerging evidence highlights that multiple interactions (or interaction domains) between p53 and FOXO transcription factors play crucial roles in positioning of both transcription factors on promoter sites, including those of p21 and p53[5]. Notably, several FOX family proteins - including FOXL2, FOXA1, and FOXO3 - can bind to the non-canonical homotypic cluster of the p53 promoter region[43]. These cooperative interactions regulate p53 transcription activation. Moreover and further emphasizing the importance of TF-interplay within promotor regions, Renault et al. have shown that p53 directly trans-activates FOXO3 by binding to a specific site in the second intron of the FOXO3 gene[44]. Interestingly, the genomic region associated with extreme longevity in humans is involved in the regulation of both p53 and FOXO3. Although FOXO3 is not essential for p53-mediated cell cycle arrest, it appears to modulate p53-dependent apoptosis[5]. Therefore, interactions between multiple TFs, including p53 and FOXOs, seem to govern the transcriptional activity of target genes. It remains to be demonstrated whether p53 and FOXO4 can synergistically regulate targeted gene transcription. Our findings indicate that the binding surface of p53/FOXO4 TADs on their DNA binding domains overlap with their respective DNA binding interface. Notably, the addition of DNA competes with p53 binding to FOXO4[FH] (Supplementary Fig. 2d) and FOXO4 binding to p53[DBD] (Supplementary Fig. 2c). This suggests that p53 and FOXO4 may form complexes within promotor regions. Furthermore, given that the p53/FOXO4 TADs are involved in self-interaction via binding to their own DNA binding domains[14,42], another regulatory mechanism could involve the formation of biomolecular condensates. Evidence supports this possibility, as recombinant p53 has been shown to phase separate in vitro[45]. It is plausible that transient interactions between multiple binding sites, including p53-p53, FOXO4-FOXO4 and p53-FOXO4, facilitate the formation of p53-FOXO4 condensates. These condensates may, in turn, play an important role in p53/FOXO4-mediated transcription, as well as other processes like PML body recruitment or the DNA-damage response. Supporting this idea, we previously observed that in senescent cells, FOXO4 and p53 localize within PML bodies and DNA-SCARS. Interestingly, the addition of FOXO4-DRI disrupts this localization, leading to the exclusion of p53 from these condensates[5]. Further research is needed to fully understand their intricate interplay and the implications for cellular fate and disease progression.

Understanding the molecular intricacies of the native interaction between p53 and FOXO4[FH] as well as the interaction of p53 and FOXO4-DRI is crucial. These insights can help optimize existing compounds and guide the development of distinct agents that disrupt the FOXO4-p53 interaction, a process shown to be pivotal for cellular senescence[5–8]. FOXO4-DRI, a cell penetrating peptide derived from FOXO4, has demonstrated success in clearing senescent cells across various mouse models by interfering with the p53-FOXO4 interaction[5]. Our findings reveal that FOXO4-DRI directly interacts with the N-terminal TAD of p53, a binding site shared with FOXO4 and several other binding partners. Interestingly, both, the TAD of p53 and FOXO4-DRI are intrinsically disordered in solution, but fold transiently upon binding, forming a complex. Notably, not only the sequence derived from the FOXO4[FH] domain but also the HIV-TAT contribute to contacts with p53[TAD2], underscoring an additional function beyond cellular uptake.

Despite being one of the most extensively studied proteins, p53 remains a challenging target for drug discovery. Its relevance in various cancer types is undeniable, yet it is considered "undruggable" or "not yet druggable"[11]. This is to a large extent due to the fact that p53 is largely disordered and lacks well-defined binding pockets for small molecules. This structural flexibility poses challenges for designing drugs that directly interact with p53. Despite immense research efforts surrounding p53 it was only recently that clinical trials specifically targeting p53 commenced[11]. Most of these approaches have in common that they target well-folded binding partners of p53, such as MDM2/MDM4, which prevents p53 degradation. Other strategies focus on reactivation of mutant p53 to restore wild-type functions, and typically target the p53 DBD. When studying the p53–FOXO4-DRI interactions using NMR spectroscopy, we made an unexpected discovery: the disordered FOXO4-DRI bound to a transiently folded α-helical region within the disordered p53 transactivation domain, termed TAD2, and triggered formation of a transiently folded tertiary structure. Optimization of the hit structure FOXO4-DRI led to a distinct class of p53-targeting molecules, out of which 2 compounds are currently in the preclinical development phase in context of oncology (WO/2023/180570, WO/2021/165538). The serendipitous discovery of FOXO4-DRI targeting the transiently folded p53[TAD2] helix provides a useful paradigm how to target other IDPs in the future.

The primary binding site for both FOXO4[FH] and FOXO4-DRI on p53 is also a binding site for several other proteins, including BRCA2, CBP/p300, HMGB1, HDM2/MDM2, Mre11, PC4, and PR/GR repeats[16–22]. As a result, when FOXO4-DRI binds, it likely competes with these other p53 regulators for the same spot. The reciprocal might also be true: other regulators may compete with FOXO4-DRI. In the future it will be important to address if such a potential competition with multiple binding partners allows p53 to be released from PML bodies and translocate to the mitochondria, ultimately inducing apoptosis.

The N-terminal transactivation domain of p53 harbors multiple phosphorylation sites that regulate its interaction with various partners, ultimately influencing p53 function[10]. So far more than 15 phosphorylation sites have been identified in response to DNA damage induced by either ionizing radiation or ultraviolet light irradiation. Notable sites include Ser15 and Ser20, which are phosphorylated for example by ATM, reducing MDM2 binding and stabilizing p53 protein levels. In a similar manner, Ser46 is phosphorylated upon DNA damage promoting p53-mediated apoptosis[46]. Under normal growth conditions, Thr55 is constitutively phosphorylated by TAF1, the largest component of transcription factor TFIID[47]. This phosphorylation promotes p53 degradation via the proteasome. In response to DNA damage, PP2A dephosphorylates Thr55, enhancing p53 stability. Our studies suggest that phosphorylation of Ser46 and Thr55 enhances the interaction between the FOXO4 FH domain and the N-terminal trans-activation domain of p53. Moreover, we discovered that phosphorylation of Ser46 and Thr55 enhanced binding to FOXO4-DRI.

Altogether, our results point towards the importance of p53 phosphorylation in cellular senescence and inhibition of p53-mediated apoptosis. However, the precise phosphorylation status of p53 in senescent cells remains elusive and understanding this regulation will be crucial for deciphering cellular senescence and disease progression.

In summary, our study sheds light on the intricate interplay between FOXO4 and p53 axis and provides important molecular insights into the mechanisms involved in maintaining senescent cell viability. By solving solution NMR structural models, we revealed that the disordered FOXO4-DRI peptide transiently binds to the disordered p53 transactivation domain. Notably, both the FOXO4-derived region and the cationic cell permeability peptide contribute to this interaction. Additionally, we demonstrate that p53 phosphorylation enhances affinity for both FOXO4 and FOXO4-DRI. These findings provide a foundation for developing p53 inhibitors to target diseases associated with cellular senescence, including cancers and other age-related

conditions. Further exploration of this dynamic interaction will be crucial for advancing therapeutic strategies and to ultimately alleviate age-related diseases related to senescent cells.

# Methods

## Plasmids

Expression constructs for full-length human p53 (Uniprot ID P04637) and shorter fragments from amino acids 1–94 (p53[1–94]), 1–94 I50A (p53[1–94 I50A]), 1–94 W53 A (p53[1–94 W53A]), 1–94 F54A (p53[1–94 F54A]), 1–94 I50A W53A F54A (p53[1–94 3A]), 14–37 (p53[TAD1]), 37–57 (p53[TAD2]), 94–312 (p53[DBD]), 1–312 (p53[1–312]), as well as expression constructs for FOXO4 fragments (Uniprot ID P98177) from amino acids 1–80 (FOXO[Nter]), 200–505 (FOXO[Cter]), 86–208 (FOXO4[FH]), and 464–505 (FOXO4[CR3]) were generated by synthesis of the corresponding optimized cDNA, for p53 or FOXO4 respectively, and insertion of these cDNA into pETM11 vector containing His$_6$-protein A-tag using NcoI/BamHI restriction sites for *Escherichia coli* (*E. coli*) expression (Genscript). Expression constructs for paramagnetic relaxation enhancement experiments were generated by site-directed mutagenesis introducing cysteine residues at following positions of FOXO4[FH]: 96, 116, 140, and 181. Expression constructs for FOXO4-LRI were generated by synthesis of the corresponding optimized cDNA based on the published sequence[5].

## Peptides

The following peptides were purchased from Peptide Specialty Laboratories GmbH: (Heidelberg, German) FOXO4-DRI, p53[TAD1] (residues 14-37), p53[TAD2] (residues 37-57), pS46 p53[TAD2], pT55 p53[TAD2], pS46/pT55 p53[TAD2], *D*-p53[TAD2], pS46 *D*-p53[TAD2], pT55 *D*-p53[TAD2], pS46/pT55 *D*-p53[TAD2], S46C p53[TAD2] (attached with a paramagnetic 3-(2-Iodoacetamido)-PROXYL tag to the cysteine), and Q52C p53[TAD2] (attached with a paramagnetic3-(2-Iodoacetamido)-PROXYL tag to the cysteine). All peptides were of the correct sequence and of >95 % purity, based on mass spectrometry and HPLC-UV.

## Oligos

CTDSP2 response element 1 DNA (sense: CCAGATAAACAACCCG, antisense: CGGGTTGTTTATCTGG) and CDKN1A-p53-RE (sense: CTCAACATGTTGGGACATGTTCCTTTT; antisense: AGGAACATGTCCC AACATGTTGAGAA) were purchased from Eurofins MWG Operon. DNA duplexes were formed by mixing equimolar amounts of sense and antisense strands in H$_2$O, heating the mixture to 95 °C for 10 min and slowly cooling to 4 °C overnight.

## Protein expression and purification

For expression of recombinant unlabeled or $^{15}$N labeled or $^{15}$N/$^{13}$C-labeled His$_6$-protein A-TEV cleavage site- tagged proteins, the different bacterial expression His$_6$-protein A-TEV cleavage site-pETM11 vectors were transformed into *Escherichia Coli* BL21-DE3 Star strain. 10 ml of liquid preculture were then transferred in 1 L lysogeny broth (LB) medium or in case of isotope labeled in minimal medium supplemented with either 6 g of unlabeled glucose (Roth) or 2 g of $^{13}$C-labeled glucose (Cambridge Isotope Laboratories) and either 3 g of unlabeled NH$_4$Cl (Roth) or 1 g of $^{15}$NH$_4$Cl (Sigma). Cells were grown to an optical density (OD) (600 nm) of 0.8 and protein expression was induced by addition of 0.5 mM IPTG followed by incubation at 20 °C for 16 hours. Cell pellets grown for expression of unlabeled, $^{15}$N-labeled or $^{15}$N/$^{13}$C-labeled disordered protein fragments (p53[1–94], p53[1–94 I50A], p53[1–94 W53A], p53[1–94 F54A], p53[1–94 3A] p53[TAD1], p53[TAD2], FOXO4[Cterm], FOXO4[Nterm], and FOXO4[CR3]) were harvested and sonicated in denaturing lysis buffer (50 mM Tris-HCl pH 7.5, 150 mM NaCl, 20 mM imidazole, 6 M urea) whereas the folded protein fragments (p53[1–312], p53[DBD], FOXO4[FH]) were harvested and sonicated in non-denaturing lysis buffer (50 mM Tris-HCl pH 7.5, 150 mM NaCl, 20 mM imidazole, 2 mM tris(2-carboxyethyl)phosphine). His$_6$-protein A-TEV cleavage site-tagged recombinant

proteins were then purified using Ni-NTA agarose (Qiagen) and the His$_6$-protein A tag was cleaved off by adding 2 (w/w) % His$_6$-tagged TEV protease for 16 h at 4 °C. After desalting using a desalting column (HiPrep 26/10, Cytiva) on an ÄKTA Pure system (GE Healthcare) to a low imidazole buffer (50 mM Tris pH 7.5, 150 mM NaCl, 20 mM imidazole, 2 mM TCEP) the untagged proteins were then isolated performing a second affinity purification using Ni-NTA beads. A final size exclusion chromatography purification step was performed in the buffer of interest on a gel filtration column (Superdex 75, Cytiva for FOXO4[FH], FOXO4[CR3], FOXO4[Cterm], p53[1–94], p53[1–312], p53[DBD], and Superdex peptide, Cytiva for p53[TAD1], and p53[TAD2]). Protein concentrations were estimated from their absorbance at 280 nm, assuming that the ε at 280 nm was equal to the theoretical ε value.

The gene encoding tagged FOXO4-LRI was similarly transformed into *Eschericia Coli* BL21-DE3 Star strain and transferred into either LB medium or isotope labeled minimal medium. After addition of 0.5 mM IPTG and expression at 20 °C for 16 h, cells were harvested and sonicated in denaturing lysis buffer (50 mM Tris-HCl pH 7.5, 150 mM NaCl, 20 mM imidazole, 6 M urea). His$_6$-protein A-tagged FOXO4-LRI lysate was then purified using Ni-NTA beads. The His$_6$-protein A tag was cleaved by adding 2 (w/w) % His$_6$-tagged TEV protease for 16 h at 4 °C and desalted to a low imidazole buffer (50 mM Tris pH 7.5, 150 mM NaCl, 20 mM imidazole, 2 mM TCEP). The protein was then subjected to a heparin column (GE Healthcare) to separate the cleaved protein from the tag by applying a salt gradient using a low salt buffer (50 mM Tris pH 7.5, 150 mM NaCl, 20 mM imidazole, 2 mM TCEP) and a high salt buffer (50 mM Tris pH 7.5, 1 M NaCl, 20 mM imidazole, 2 mM TCEP). To purify the cleaved FOXO4-LRI and to remove the uncleaved protein a final gel filtration step using the Superdex peptide column (GE Healthcare) into the final measurement buffer (50 mM NaH$_2$PO$_4$/Na$_2$HPO$_4$ (pH 7.5), 0.04% NaN$_3$)) was performed.

## Stable cell line production

RPE-1 p53KO cells were transduced by viruses expressing p53-3HA-GFP-miniturbo and p53-Δtad2-3HA-GFP-miniturbo. Then, cells were selected by blasticidin for 10 days. Cells, which were expressing 3HA-GFP-miniturbo, p53-3HA-GFP-miniturbo-p53 and p53-Δtad2-3HA-GFP-miniturbo were sorted by FACS using GFP (Supplementary Fig. 17).

## Cell culture

p53 mutant human TERT-immortalized retinal pigment epithelial 1 (RPE1) cells (RPE1 TP53$^{-/-}$; denoted RPE-1 p53KO) cells were kindly provided by Joanna Loizou (CeMM Research Center for Molecular Medicine of the Austrian Academy of Sciences, Vienna, Austria)[48]. Stable RPE-1 p53KO cells expressing p53-3HA-GFP-miniturbo and p53-ΔTAD2-3HA-GFP-miniturbo and RPE-1 p53KO cells were cultivated in DMEM containing 4.5 g/L glucose, 2 mM glutamine, 10% FBS, 1% Penicillin/Streptamycin and 5 μg/mL Blasticidin at 37 °C, 5% CO$_2$ in humidified atmosphere. For the preparation of cell lysates for pull down experiments, cells were grown to confluence and incubated with 10 μM Etoposide (Sigma) for 24 h. Cells were harvested in ice cold PBS pH 7.4 and lysed in lysis buffer (10 mM Tris pH 7.4, 1% IGEPAL®CA-630, 0.1% SDS, 140 mM NaCl, 2 mM MgCl$_2$) supplemented with protease inhibitor cocktail, Benzonase (Merck Millipore) and RNase (molecular biology, Thermo Fisher). Samples were incubated on ice for 20 min, lysed by sonication and centrifuged for 15 min at 24,104 g and 4 °C. Pellets were discarded. Protein amount of lysates was determined by BCA protein assay. Lysates were then diluted to a protein concentration of 1 mg/mL in ice-cold dilution buffer (10 mM Tris pH 7.4, 1% IGEPAL®CA-630, 0.1% SDS, 140 mM NaCl, 2 mM EDTA) supplemented with protease inhibitor cocktail. Due to unequal expression levels of p53 wt and p53 ΔTAD2, lysates containing p53 ΔTAD2 were diluted up to 4-fold in RPE-1 p53KO-lysate.

## Pull-down assay

Streptavidin agarose beads (ThermoFisher) were washed with Tris-buffered saline (TBS; 10 mM Tris, 140 mM NaCl) pH 7.4 and were then incubated with 125 µM biotinylated peptide CLO4201 in TBS pH 7.4 or 26 µM biotinylated FOXO-FH in TBS pH 8.0 at 4 °C for 1 h. After blocking of unoccupied binding sites with 250 µM D-(+)-Biotin (Sigma) in TBS pH 7.4 for 5 min at RT beads were washed 3 times with binding buffer (10 mM Tris pH 7.4, 1% IGEPAL®CA-630, 0.1% SDS, 140 mM NaCl, 1 mM EDTA) and were subsequently incubated with 1 mg/mL cell lysate for 2.5 h at 4 °C and constant shaking. Beads were then washed 3 times with 10-fold bead volume of binding buffer, 3 times with washing buffer A (10 mM Tris pH 7.4, 0.5% IGEPAL®CA-630, 0.05% SDS, 320 mM NaCl, 1 mM EDTA), and 3 times with washing buffer B (10 mM Tris pH 7.4, 500 mM NaCl) at RT. After a final washing step with TBS pH 7.4 bound protein was eluted by boiling the beads in SDS-sample buffer containing 6% β-mercaptoethanol for 10 min. Samples were subjected to SDS-Page and Western blotting. p53 polyclonal antibody (rabbit, 1:3000, Cat # 10442-1-AP, Lot # 00135720, Proteintech) was used as primary antibody and goat-anti rabbit HRP conjugate (goat, 1:10000, Cat # 1858415, Lot # HG106488, Pierce) as secondary antibody (Supplementary Fig. 16).

## Paramagnetic labelling

Mutant FOXO4$^{FH}$ expression constructs were designed, in which the amino-acids at positions 96, 116, 140, and 181) were mutated to cysteines (*vide infra*). These constructs allowed the linkage of the paramagnetic tag at mutated sites. Purified FOXO4$^{FH}$ cysteine versions were desalted on Hiprep 26/10 Desalting column (Cytiva) in a 1 M Tris-HCl buffer at pH 8 and incubated 16 hours at 4 °C in presence of 8 times stoichiometric excess of the paramagnetic 3-(2-Iodoacetamido)-PROXYL tag (Sigma). The excess of the paramagnetic label was removed using a gel filtration step (Superdex 75, GE Healthcare) into the final measurement buffer (50 mM NaH$_2$PO$_4$/Na$_2$HPO$_4$, pH 7.5, 0.04 % NaN$_3$)

## Isothermal titration calorimetry (ITC)

All proteins were equilibrated either in a buffer containing 20 mM Hepes (pH 7.0), 50 mM NaCl and 2 mM tris(2-carboxyethyl)phosphine (TCEP). ITC measurements were taken with a MicroCal VP-ITC instrument (Microcal, Northhampton, USA) with 18 rounds of 15 µl injections at 25 °C. Integration of peaks corresponding to each injection, subtraction of the contribution of protein dilution, correction for the baseline and curve fitting were performed using the MicroCal VP-ITC analysis software provided by the manufacturer. Curve fitting was done using standard one-site model and gives the equilibrium binding constant (K$_a$), the stoichiometry (n), entropy (ΔS), and enthalpy of the complex formation (ΔH).

## NMR spectroscopy

All NMR spectra were recorded at 298 K either on a 700 MHz Bruker Avance III NMR spectrometer equipped with a TCI cryoprobe or on a 600 MHz Bruker Avance Neo NMR spectrometer equipped with a TXI 600S3 probehead at 298 K. All samples were in 50 mM NaH$_2$PO$_4$/Na$_2$HPO$_4$, pH 7.5 and 10 % D$_2$O was added for the lock. Chemical shifts of the recombinant FOXO4-LRI were assigned using the following three-dimensional spectra: HNCACB, CBCA(CO)NH, (HN)N(CA)NNH, HN(NCA)NNH and (H)CC(CO)NH. The assignment of FOXO4$^{FH}$, FOXO4$^{CR3}$, p53$^{TAD2}$, and p53$^{1-94}$ has been described previously[42] and had already been submitted to the BMRB under the accession numbers 50398, 50402, 51125, and 51124, respectively. Resonance assignments of FOXO4-LRI were submitted to the BRMB under the accession number 52458 (Supplementary Fig. 15).

$^{13}$C/$^{15}$N filtered, $^{13}$C-edited- and $^{13}$C/$^{15}$N filtered $^{15}$N-edited 3D NOESY-HSQC NMR experiments of the FOXO4$^{FH}$ – p53$^{TAD2}$ complexes have been recorded with a mixing time of 200 ms for a sample composed of 516 µM $^{13}$C,$^{15}$N-labeled FOXO4$^{FH}$ in presence of 623 µM unlabeled p53$^{TAD2}$ and for a sample composed of 800 µM $^{13}$C,$^{15}$N-labeled p53$^{TAD2}$ in presence of 1000 µM unlabeled FOXO4$^{FH}$.

To obtain a structural model of the FOXO4-DRI–p53$^{TAD2}$ complex, we recorded $^{13}$C,$^{15}$N filtered, $^{13}$C-edited 3D NOESY-HSQC experiments with a mixing time of 200 ms on a sample composed of 1160 µM $^{13}$C,$^{15}$N-labeled p53$^{TAD2}$ in presence of 1440 µM FOXO4-DRI and on the mirror image complex sample composed of 850 µM $^{13}$C,$^{15}$N-labeled recombinant FOXO4-LRI in presence of 1080 µM unlabeled *D*-p53$^{TAD2}$. Exemplary NOE strips are shown in Supplementary Fig. 13. Furthermore, we recorded 2D NOESY spectra with $^{13}$C,$^{15}$N filtering in F1 and/or F2 with 200 ms mixing times.

NMR relaxation experiments for p53$^{TAD}$ were recorded using $^{13}$C,$^{15}$N isotope-labeled samples at 300 µM in a buffer containing 50 mM NaH$_2$PO$_4$/Na$_2$HPO$_4$, pH 7.5, 0.04 % NaN$_3$, 2.5 mM DTT and 10 % D$_2$O for the lock in absence and in presence of 400 µM unlabeled FOXO4$^{FH}$ or 400 µM synthetic FOXO4-DRI peptide, respectively. $^{15}$N $R_1$ relaxation data was recorded using the HSQCT1ETF3GPSI3D.2 Bruker pulse sequence (8 scans, 256 points in F1, 1024 points in F2, 9615.410 Hz spectral width in F1, 1825.051 Hz spectral width in F2), using a variable delay of 0.010000, 2.000000, 0.030778, 1.272627, 0.063183, 0.806249, 0.113724, 0.507217, 0.192548, 0.507217, and 0.315484 s. $^{15}$N $R_{1\rho}$ relaxation data was recorded using the HSQCTRETF3GPSI3D Bruker pulse sequence (16 scans, 256 points in F1, 1024 points in F2, 9615.410 Hz spectral width in F1, 1825.051 Hz spectral width in F2), using a variable delay of 0.010000, 0.200000, 0.011984, 0.130552, 0.015078, 0.086024, 0.019903, 0.057473, 0.027429, 0.057473, and 0.039167 s. $^{15}$N{$^1$H} heteronuclear NOE data was recorded using the HSQCNOEF3GPSI bruker pulse sequence (32 scans, 256 points in F1, 1024 points in F2, 1278.772 Hz spectral width in F1, 9615.385 Hz spectral width in F2) using a three seconds saturation period.

NMR relaxation experiments for the recombinant *L*-version of the FOXO4-DRI sequence were recorded using $^{13}$C,$^{15}$N-labeled samples at 300 µM in a buffer containing 50 mM NaH$_2$PO$_4$/Na$_2$HPO$_4$, pH 6.5, 0.04 % NaN$_3$, 2.5 mM DTT and 10 % D$_2$O in the absence and in presence of 400 µM synthetic *D*-p53$^{TAD2}$ peptide using the same NMR parameters.

Spectra were processed using TOPSPIN 4.1 (Bruker Biospin, Rheinstetten, Germany) and analyzed using CcpNmr 2.5[49].

## NMR chemical shift perturbation analysis

The CSPs were calculated according to the following equation:

$$CSP = \sqrt{(\delta H)^2 + \frac{(\Delta \delta N)^2}{7}} \tag{1}$$

where $\Delta \delta H$ and $\Delta \delta N$ indicate the chemical shift changes of the amide proton and nitrogen, respectively, using CcpNMR 2.5[49].

## NMR relaxation analysis

Data analysis von $R_1$ and $R_{1\rho}$ measurements were performed using CcpNMR 2.5[49]. As fitting function, the following equation was used

$$M_z(t) = M_0^{-R_i t} + C \tag{2}$$

where $M_z(t)$ is the longitudinal relaxation at the recovery delay time $t$, $M_0$ is the equilibrium magnetization and $R_i$ is either the longitudinal relaxation rate $R_1$ or $R_{1\rho}$. The transverse relaxation in the rotating frame $R_{1\rho}$ was then converted to $R_2$ as described previously[50,51] using the formula

$$R_{1\rho} = R_1 \cos^2\theta + R_2 \sin^2\theta \tag{3}$$

in which $\theta = arctan(1/\Omega)$ is the tilt angle between the static magnetic field and the effective field in the rotating frame.

## Structural model determination

Due to intermediate exchange only a few inter-molecular NOEs could be obtained. Of the 119/48 total intermolecular NOEs for p53$^{TAD2}$-FOXO4-DRI/p53$^{TAD2}$-FOXO4$^{FH}$ complexes only 19/2 NOEs could be assigned unambiguously. To obtain additional restraints for structural model determination of the p53$^{TAD2}$-FOXO4$^{FH}$ complex, we determined paramagnetic relaxation enhancements (PREs) for a set of 7 complexes containing spin-labeled (SL) samples ($^{15}$N-labeled FOXO4$^{FH}$ + S46C-SL p53$^{TAD2}$, $^{15}$N-labeled FOXO4$^{FH}$ + Q52C-SL p53$^{TAD2}$, $^{15}$N-labeled p53$^{TAD2}$ + Q112C-SL FOXO4$^{FH}$, $^{15}$N-labeled p53$^{TAD2}$ + D140C-SL FOXO4$^{FH}$, $^{15}$N-labeled p53$^{TAD2}$ + N145C-SL FOXO4$^{FH}$, $^{15}$N-labeled p53$^{TAD2}$ + S159C-SL FOXO4$^{FH}$, $^{15}$N-labeled p53$^{TAD2}$ + S162C-SL FOXO4$^{FH}$). To measure the effect of the paramagnetic label, $^{1}$H,$^{15}$N and $^{1}$H,$^{13}$C HSQC spectra with long inter-scan delays (3 s) of 100 μM $^{13}$C/$^{15}$N-labeled p53$^{TAD2}$ in presence of one stoichiometric equivalent paramagnetically labeled FOXO4$^{FH}$ were recorded before and after addition of 5 mM ascorbic acid. To obtain distance restraints of paramagnetic p53$^{TAD2}$ towards FOXO4$^{FH}$, were performed using p53$^{TAD2}$ peptides which are labeled at position 46 and position 52 respectively. Those peptides were incubated with one stoichiometric equivalent with 100 μM $^{13}$C,$^{15}$N-labeled FOXO4$^{FH}$.

Intensity ratios between the paramagnetic and diamagnetic states were computed for all assigned and non-overlapping $^{1}$H, $^{15}$N, and $^{1}$H, $^{13}$C HSQC cross-peaks, employing a correlation time of the electron-spin interaction vector of 4 ns[52,53]. Inter-molecular PRE restraints were incorporated in the structure calculations, considering four non-interacting copies of the spin label to accommodate varying spin label orientations.

Structural models were calculated using modified CNS protocols in the ARIA/CNS setup[53,54], similar as described for the FOXO4$^{FH}$-FOXO4$^{CR3}$ structure determination pipelne[42]. We generated 100 structural models of the FOXO4$^{FH}$ bound to the p53$^{TAD2}$, starting with the 3D structure of FOXO4$^{FH}$ (PDB: 1E17 [55]) and the p53$^{TAD2}$ domain as an elongated chain. In brief, the protocol consists of the following steps: (1) generation of flexible regions in FOXO4$^{FH}$ (residues 86-104 and 186-207), the p53$^{TAD2}$ peptide, and spin labels, randomization of the flexible regions; (2) molecular dynamics simulated annealing restraining FOXO4$^{FH}$ (without flexible regions) harmonically to its template structure (using a non-crystallographic force constant of 10,000 kcal mol$^{-1}$Å$^{-2}$), with additional dihedral angle restraints from $^{1}$H, $^{15}$N, and $^{13}$C chemical shifts using TALOS-N[56], hydrogen bond restraints (region p53 residues 48-55; based on consistent formation of α-helical structure in initial rounds of structure calculations in which no hydrogen bond restraints were defined), inter-molecular PRE restraints (4 non-interacting copies of the spin label to account for different spin label orientations), inter- and intra-molecular NOE restraints, and ambiguous distance restraints for residues showing CSPs upon binding. Of the 48 inter-molecular NOE restraints, only 2 NOEs could be assigned unambiguously and correspond to contacts between p53-Trp53 HE# - FOXO4-Leu160 HD# and p53-Trp53 HD#−FOXO4-Leu109 HD# (Supplementary Note 1). The 10 structural models with the lowest associated energy from 100 calculated structural models were used to prepare figures using the PyMOL Molecular Graphics System, Version 2.4 (Schrödinger, LLC).

For the complex structural model of FOXO4-DRI and p53$^{TAD2}$ we generated 1000 structural models of FOXO4-DRI bound to p53$^{TAD2}$. Both molecules were started as an elongated chain. For FOXO4-DRI we used D-amino acids. The restraints were derived from (i) NOE restraints from intermolecular and intramolecular cross-peaks in NOESY spectra (described in NMR spectroscopy section), (ii) TALOS derived p53$^{TAD2}$ and FOXO4-DRI backbone dihedral angles, (iii) hydrogen bond restraints (regions p53 residues 47−52, FOXO4-DRI residues 9−18; based on consistent formation of α-helical structure in initial rounds of

structure calculations in which no hydrogen bond restraints were defined). Of the 119 inter-molecular NOE restraints, only 20 NOEs could be assigned unambiguously (Supplementary Note 2). Ambiguous restraints to Arg residues in the HIV-TAT have been excluded given the flexibility of this region. The 10 structural models with the lowest associated energy from 1000 calculated structural models were used to prepare figures using the PyMOL Molecular Graphics System, Version 2.4 (Schrödinger, LLC).

## Molecular dynamics simulations

Molecular dynamics simulations have been performed using the AMBER ff19SB force field[57] and the solvent molecules were represented as OPC water molecules[58]. No experimental restraints were used for the MD simulations. We used the force field parameters for single-protonated phosphorylated Ser (S1P) and Thr (T1P) residues developed by Sticht and co-workers[59]. Periodic boundary conditions were used with a cutoff of 10 Å for non-bonded interactions and the Particle Mesh Ewald method for long-range electrostatics. The proteins were embedded in a box of OPC water molecules (minimum distance of 12 Å from the solute to the boundaries of the box) and the net charge of the system was set to zero by neutralization with Cl-ions using the program tLEaP included in AmberTools21 (https://ambermd.org)[60].

The system underwent a three-stage minimization process: first for protons, followed by solvent molecules and counter ions, and finally, the entire system. Subsequently, each system underwent independent simulations lasting 500 ns each (totaling 1.5 μs per system) at 298 K, employing the Langevin thermostat with a time step of 0.2 fs (Supplementary Fig. 14 and Supplementary Table 1). Harmonic restraints of 200 kcal mol$^{-1}$Å$^{-2}$ were applied to the remaining atoms at each stage. Thermal equilibration was then conducted for 20 ps, gradually increasing the temperature from 100 K to 298 K using the Langevin thermostat (collision frequency of 1 ps$^{-1}$). The harmonic restraints were released in five steps, transitioning the system from an NVT to an NPT ensemble over 100 ps. Subsequently, each system underwent independent simulations lasting 500 ns each (totaling 1.5 μs per system) at 298 K, employing the Langevin thermostat with a time step of 0.2 fs. SHAKE algorithm(DOI:10.1016/0021-9991(77)90098-5) was applied to constrain covalent bonds involving hydrogen atoms. Trajectory analysis of the molecular dynamics simulations was carried out using the cpptraj program. The major conformational clusters from each of the MD simulations were obtained with the graph-based program BitQt[61]. A cut-off of 2 Å in the backbone was applied for delimiting the solutions belonging to the same cluster. Ramachandran plots were plotted using the matplotlib and MDAnalysis libraries in python3. MD data has been submitted to Zenodo (https://doi.org/10.5281/zenodo.10963887).

## Markov state model

The conformational space of p53$^{TAD2}$ was analyzed via Markov State Model (MSM)[62]. 1500 frames from each MD trajectory of each simulated system (1.5 μs per system) were printed with cpptraj and considered for analysis. In all cases, water molecules, counter ions, and binding partners (FOXO4-DRI, FOXO4$^{FH}$) were removed before analysis. First, the number of dimensions were reduced using time-structure independent component analysis (tICA)[63,64]. As features, we selected the backbone torsions of p53$^{TAD2}$. Then, the MD simulations was divided into 85 microstates using the clustering algorithm k-means. A lag-time of 5 ns was defined for all systems. The MSM models were validated using the Chapman-Kolmogorov test[65] with with a confidence of 95%. Finally, the metastable sets were computed using the PCCA+ spectral clustering implemented in pyEMMA. As outcome, we computed the free energy surfaces and transition probabilities. The entire analysis was performed using pyEMMA[66] freely available at github (https://github.com/markovmodel/PyEMMA).

## Reporting summary

Further information on research design is available in the Nature Portfolio Reporting Summary linked to this article.

## Data availability

The MD data generated in this study have been deposited in the Zenodo repository under the accession code 10963887. Resonance assignments have been deposited to the BMRB under the accession codes 52458 (FOXO4-LRI) and 52457 (p53TAD2 bound to FOXO4-DRI). Additional NMR data are included in the Source Data file. BMRB codes of previously published resonance assignments used in this study are 50398 (human FOXO4 FH domain), 50402 (human FOXO4 CR3), 51125 (human p53 TAD2), 51124 (human p53 1–94), and 51753 (human p53 DBD). PDB codes of previously published structures used in this study are 4HJE (human p53 DBD), 3L2C (human FOXO4 FH domain bound to DNA) and 1E17 (human FOXO4 FH domain). Source Data are provided in Source Data. All other relevant data are available from the corresponding author on request. Source data are provided with this paper.

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

## Acknowledgements

D.P.-R. and P.A.S.-M. thank for MedBioNode (Med Uni Graz, Austria) computation time. We thank Joanna Loizou and Anna Schrempf for providing p53 mutant human TERT-immortalized retinal pigment epithelial 1 (RPE1) cells. We thank Christopher Koscher for his assistance with protein expression and purification and Florian Grebien for help with establishing the RPE-1 cell lines stably expressing p53 and p53 mutants. We thank Martina Mairold and Dorian Gorkievicz for their excellent support in culturing RPE-1 cells and performing pull-down assays including Western Blot analysis. T.M. is grateful to the Austrian Science Fund (FWF) for excellence cluster 10.55776/COE14, Grants DOI 10.55776/P28854, 10.55776/I3792, 10.55776/DOC130, and 10.55776/W1226, the Austrian Research Promotion Agency (FFG) grants 864690 and 870454; the Integrative Metabolism Research Center Graz; the Austrian Infrastructure Program 2016/2017; the Styrian Government (Zukunftsfonds, doc.fund program); the City of Graz; and BioTechMed-Graz (flagship project). This project was funded in part by the FFG and the European Union (EFRE) under grant 912192. E.S. and D.P.-R. were trained within frame of the PhD program Molecular Medicine. QZ and SU were trained within of the PhD program Metabolic and Cardiovascular Disease (DK-MCD) and Biomolecular Structures and Interactions (BioMolStruct). For open access purposes, the author has applied a CC BY public copyright license to any author accepted manuscript version arising from this submission.

## Author contributions

B.B. methodology, investigation, formal analysis, writing—original draft, writing—review & editing. E.S. methodology, investigation, formal analysis, writing—original draft, writing—review & editing. D.P.-R. methodology, investigation, formal analysis, writing—review & editing. M.P. methodology, investigation, formal analysis, writing—review & editing. Q.Z. methodology, investigation, formal analysis, writing—review & editing. S.U. methodology, investigation, formal analysis, writing—review & editing. P.L.J.d.K. conceptualization, resources, writing—review & editing, funding acquisition. B.M.T.B. resources, writing—review & editing, supervision. P.A.S.-M. investigation, formal analysis, resources, writing—review & editing, supervision. T.M. conceptualization, methodology, investigation, formal analysis, resources, writing—original draft, writing—review & editing, funding acquisition, supervision

## Competing interests

We disclose the following potential conflicts of interest regarding the publication of this manuscript: This research was partly funded by Cleara Biotech. P.D.K. and T.M. hold shares in Cleara Biotech. B.B., E.S.,

P.D.K., and T.M. hold a published patent (WO/2021/165538) that describes compounds, wherein said compounds bind to p53 or preferably bind to p53 and inhibit the interaction of FOXO4 with p53 in a cell. The binding and structural studies of FOXO4-DRI with p53 which are reported in this manuscript are related to the patent. P.D.K. and T.M. hold a published patent (WO/2023/180570) that covers the in vitro method for identifying anti-senescence compounds based on detecting the binding of said compound in the presence of at least one phosphorylated amino acid in the transcription activation domain (TAD) domain of mammalian protein p53. The remaining authors declare no competing interests.
