## [Transparent Peer Review file · Nature Communications]

The Disordered p53 Transactivation Domain is the Target of FOXO4 and the Senolytic Compound FOXO4-DRI

Corresponding Author: Professor Tobias Madl

Version 0:

Reviewer comments:

Reviewer #1

(Remarks to the Author)

In this manuscript the authors describe in detail how the two important proteins p53 and FOXO4 interact. They further corroborate earlier data that the transactivation domain 2 is the primary interaction site on p53 and identify the FH domain as the reciprocal site on FOXO4. Structural models of the p53-TAD2 FOXO4-FH complex are generated by combining NMR data and molecular dynamics simulations, which reveal partial folding of the unstructured domains upon complex formation. The authors resort to a multitude of data sources like relaxation data, Ca chemical shifts etc. and combine the findings to a plausible model. Further, they examine the influence of phosphorylation on the affinity of the interaction and identify pT55 as a phosphorylation site that enhances the affinity.

On top of characterizing the natural interaction, they also determine details of binding of the potentially therapeutic peptide FOXO4-DRI, which is important for developing peptide-based drugs against cellular senescence.

Overall, the level of characterization of this difficult system is impressive. The paper is well written and understandable, and the results and conclusions are sound, although I had hoped for one or the other additional experiment.

The authors for example state that the interaction of p53-TAD2 and FOXO4-FH is mediated by electrostatic and hydrophobic interactions. Was the salt dependence of the affinity measured? Or could key hydrophobic residues be mutated? This would add experimental evidence for this statement. Otherwise, I would rephrase the text that there are strong indications for hydrophobic and electrostatic nature of the interaction.

In the same paragraph, it is stated that DNA and p53-TAD2 have overlapping binding surfaces on FOXO. Would it be a large effort to check if there is actual competition in an experiment?

Also, the authors state that the FOXO4-DRI peptide disrupts the interaction between FOXO4-FH and p53-TAD based on chemical shift perturbations. Could a competition experiment be performed that shows that FOXO4-FH indeed becomes free in presence of FOXO4-DRI? This would dismiss potential doubts about allosteric effects, that CSPs generally can suffer from. While I understand that DNA binding to FOXO4-FH is less in the focus of this manuscript, the competition of FOXO4-DRI and FOXO-FH definitely is, and therefore I would think that this experiment is important for this paper, especially given that the required resources all seem to be present.

The data on phosphorylation are nice and convincing. Can the authors speculate on the influence of pSer46 on the alpha helicity of the subsequent residues?

The affinity of FOXO4-DRI to p53(1-94) is stated to be 0.4 μ M. (5-fold "higher" than FOXO4-FH. Is that stronger or weaker? I was under the impression that DRI binds more strongly.) However, all later ITC measurements indicate 10–100 fold weaker affinity, and NMR spectra appear to be in fast exchange, which would not support a nanomolar K_d. This seeming disparity should be discussed in the paper.

The structure reveals several insights into the interaction. Here I have a more general remark: is it the right approach to chase down the ONE lowest energy conformation of such a complex? It reminds me of the controversy around chaperone-

client interactions, where it was tried to find the preferred binding mode of a client protein, although the better representation is certainly several different binding modes of similar binding energy. I suspect that this interaction might also have this characteristic, in the sense that there are several conformations that are close in binding energy. The presented conformation was “artificially” enriched by calculating 1000 conformers instead of the 100 usual, hydrogen bonds were enforced, and the MD started only from the lowest energy structure. Is there a way to represent this complex with several favorable conformations? I suspect their relative binding energies are not too far apart from each other. Maybe the authors can discuss whether their data rather indicates that there is a single conformation of the complex as “underlying truth” and that the low density/low accuracy of restraints lead to a somewhat fuzzy model, or that the determined structure is one preferred conformation among several interaction modes that have similar energy.

The manuscript should only be accepted upon reception of the BMRB codes. However, I'm very happy to see the submission files and the assignments in user-friendly Xeasy format.

There are a number of typos in the manuscript. Here are a few:

“Aging“ and “ageing” are both used, e.g. lines 36 and 38. Stick to one.

Line 66: “coping” instead of copying

Line 140: there might be a mix up between labeled and unlabeled proteins.

Line 301: comma missing after “summarizing”

Line 427: were affected “more strongly”

Line 438: “we” conducted...

Line 513: “efforts” instead of affords

Overall, I'd like to congratulate the authors for a nice piece of work which certainly deepens our insights into the workings of FOXO4 and p53, and showing potential avenues for future treatment of senescence.

Reviewer #2

(Remarks to the Author)

In the ms, the authors mainly present the solution NMR structure complex of p53 with FOXO4 (forkhead or FH domain) or FOXO4-DRI, with FOXO4 the Forkhead box protein O 4 or a transcription factor (TF) of FOX family, and FOXO4-DRI a disordered cell-penetrating peptide as a drug molecule to target on p53 interaction.

Although the authors provided fairly thorough background information about the studied systems, some basic questions were not explained nor brought about to attention. For example, what is the molecular role of the FOXO4-p53 interaction, e.g., in regard to FOXO4-DNA and p53-DNA interactions or regulation functions, respectively? Since FOXO4 is suggested to particularly interact with p53-TAD2, while TAD2 is supposed to interact with the DBD (DBD of p53 or DBD of FOXO4 or other TFs? on page 3 line 85), can one expect that FOXO4 interferes with p53 promoter recognition? Any further literature evidences? Similarly, how is the TF function of FOXO4 impacted by its interaction with p53? Btw, is the FH domain also the DBD of FOXO4?

The authors later show that “The interface on FOXO4FH covered by p53TAD2 overlaps with the one bound to DNA indicating a competition between the binding events”. Are there additional experimental or literature evidences supporting that DNA and p53-TAD2 compete for the FOXO-FH binding?

On page 8-9, the authors address “the FOXO4FH-p53TAD2 complex structure reveals overlapping binding surfaces with DNA”, “Fig 3a shows an overlay of the top 10 lowest-energy models”, and with extensive MD simulations” conducted. How were the energy models generated? The authors should provide full technical descriptions and procedures to justify the modeling part, and provide structural models and files for future validation. Similarly, how about the FOXO4-DRI-p53 interaction models?

For the MD simulations, it was not clear how extensive or how long (and how many) the simulations were conducted, whether the systems were well equilibrated (e.g., by showing RMSDs overall simulation time), or the interaction interfaces were stabilized, or say, the results shown in Fig 3 converged or not. Similar concerns apply also to the FOXO4-DRI-p53 simulations.

(Remarks to the Author)

The manuscript "The disordered p53 transactivation domain is the target of FOXO4 and the senolytic compound FOXO4-DRI" by Bourgeois et al. describes a structural and biophysical characterization of the interaction between the p53 N-terminal region 1-94 (including both TAD domains) and the Forkhead domain of FOXO4 (FOXO4-FH). This manuscript is a structural follow-up on a seminal paper from 2016, where the authors could show that perturbing the FOXO4-p53 interaction induces apoptosis of senescent cells. The current manuscript aims to provide a structural basis of this phenomenon using purely in vitro-based biophysics and structural biology. The regions involved in this interaction were determined by NMR chemical shift perturbations (CSPs) and a structural model of FOXO4-FH with p53-TAD2 was provided based on NOEs, dihedral angles, CSPs, paramagnetic relaxation enhancements (PREs) with additional support from MD simulations. Similarly, they could demonstrate how the FOXO4-DRI peptide discovered earlier binds to p53-TAD2 by NMR spectroscopy. This interaction would compete with FOXO4 for p53 binding and has an effect on senescent cells.

While this is an impactful interaction and detailed knowledge on this is important, the study misses a thorough validation of the structural findings in form of a mutational analysis. This is surprising, as the authors seem to have the means to perform these experiments. This would be done preferably in vivo or at least in cellular assays, if possible, to show that the identified residues are indeed important. This is especially important, as the interaction is based on rather unspecific charge-charge interactions, which altogether seem to be rather weak, as all NMR titration experiments exhibit weak CSPs in the fast exchange regime (Figure 1 and related Supplementary Figures). This is the main major concern I have with this study. Although an importance of the FOXO4-p53 has been shown and this seems to be the only regions identified, the interaction seems so weak that it is hard to fathom how this will compete with all the other proteins and DNA, p53 and FOXO4 interact with. The same region of FOXO4 interacts with DNA, which is probably much stronger than with p53-TAD2 (hard to say, as the authors do not provide affinities for their FOXO4-p53-TAD2 interaction, although the data for that seems to be present in the NMR titrations). Also, the FOXO4-CR3 region interacts at the same spot with its own FOXO4-FH domain, which is covalently attached and therefore not diffusion-limited.

One possibility where this weak interaction could be of relevance is in biomolecular condensates. This interaction seems to occur in PML bodies, which, to my knowledge are biomolecular condensates. However, the authors do not consider this in their introduction or discussion, although it could be a neat explanation. There are reports about p53 condensation properties. Nevertheless, a mutational analysis is needed to provide a validation of their structure. Are the identified residues known as disease-related mutations? In the meanwhile, there should be enough cancer omics data that even rare disease-mutations can be compared.

Another major concern is the presentation of the NMR data. I cannot find any statistics table for the NMR structures. The structures are not submitted to the PDB and to my knowledge this is required by the journal. It is difficult with structures based on integrative modelling, but here a classical NMR approach is used supplemented with PREs. The authors claim to have included 48 intermolecular NOEs in the structure calculation, but they are nowhere shown also not in the provided BMRB files (they include only the chemical shift assignments). For such a small interaction surface, 48 intermolecular NOEs are a lot, and they could have been already enough for obtaining an NMR structure ensemble? Some exemplary NOE strips in the supplement would really strengthen the manuscript. In the results part, it is somewhat mentioned in passing that due to intermediate exchange and extensive line broadening only a few NOE-based distance restraints were obtained. Again, 48 intermolecular NOEs are not a few. At least in the methods section it says 48 were included in the structure calculation. Also, the NMR titrations in Figure 1 are clear fast exchange to me, at least from the few titration points shown. From the KDs determined by ITC between p53-1-94 and FOXO4-FH, intermediate line broadening could be expected, but ITC data for FOXO4-FH and p53-TAD2 is missing (it should be provided). There is also only an ITC table and no data shown as opposed to the phosphorylation experiments, where the ITC data is shown in the supplement (but NOT in the table). The authors acquired PRE data. There is quite a bit of confusion with the PREs. While supplementary figure shows four FOXO4-FH variants with engineered spin label sites, in the methods section (structure determination) it says that 5 are included in the calculation. Also, the mutated residues are different from the figure, "structure determination" method section and in the "paramagnetic labelling" section. The PRE plots do not have error bars. At least the authors should show the HSQCs in the para- and diamagnetic state and some zooms to certain residues to estimate how good and reliable the data is.

To me it seems that the effects are stronger for the p53_1-94 than for the single TADs, as if both TAD bind stronger (multivalency?). However, in the reversed titration this does not seem to be the case.

The sentence on line 229: However, TAD2 did not exhibit secondary structure elements in its free state but does so in presence of FOXO4-FH. This is a bit of an overstatement. In Figure S3A, I see three black and three orange positive bars with still very low delta values. And these three are interrupted by one (orange) or two (black) residues not exhibiting helical secondary chemical shifts. Altogether it has to be made clearer that this interaction is very transient and that it is an ensemble view and that the off-rate is possibly very high (rotational tumbling is not affected as shown by the authors). The only part where this is mentioned in passing is on lines 267-270.

Why is the interaction of phosphorylated p53-TAD2 with FOXO4-FH and FOXO4-DRI spontaneously exothermic? (Figure S6).

Minor concerns:

Altogether, it seems that the manuscript was prepared in a rush and it would need considerable language editing and proof-reading. A few examples:

1. Line 51: "cells cells"
2. Line 54: "including for example".
3. Line 66: Copying should be coping
4. Line 94: FOXO4 is member, an "a" is missing and "the" afterwards.
5. Line 98: "such as for example"
6. Line 206: "vide infra", why not just saying "see below"
7. Line 404: "what" is misplaced somehow?
8. Line 407: immense importance?
9. Line 513: affords

Panels in Figure 1 are not aligned very well (width of spectra and plots).

Version 1:

Reviewer comments:

Reviewer #1

(Remarks to the Author)

The authors have addressed essentially all the concerns I had with refined or new experiments, and the new results largely support their hypotheses.

Reviewer #2

(Remarks to the Author)

pls find the file of my comments uploaded.

[Editorial Note: This attachment is displayed at the end of the file]

Reviewer #3

(Remarks to the Author)

The authors have erased most of my concerns but two new ones have come up. However, I think this should be possible to resolve in short order. Therefore, I recommend acceptance of this manuscript for publication at Nature Communications provided my two remaining concerns are satisfyingly addressed:

1. the new ITC data (mutations of hydrophobic residues compared with wild type) do not show a large enough difference that makes a single experiment convincing enough. It seems the error bars are from the fit and not from triplicates. It needs to be shown with these small difference if they are not because of experimental variation.
2. The explanation for the difference in affinity of ITC data and the fast exchange regime in NMR titrations, indicating weaker binding is not satisfying (different buffers). The ITC buffer actually has more salt and would thus result in a weaker affinity? Can an explanation be found in the nature of the experiments?

Version 2:

Reviewer comments:

Reviewer #2

(Remarks to the Author)

The authors have addressed my concerns and incorporated revisions. I'd recommend it for publication.

Reviewer #3

(Remarks to the Author)

The authors have addressed my remaining concerns with additional experiments satisfyingly.

REVIEWER COMMENTS

Comments are highlighted with a green background

Reviewer #1 (Remarks to the Author):

In this manuscript the authors describe in detail how the two important proteins p53 and FOXO4 interact. They further corroborate earlier data that the transactivation domain 2 is the primary interaction site on p53 and identify the FH domain as the reciprocal site on FOXO4. Structural models of the p53-TAD2 FOXO4-FH complex are generated by combining NMR data and molecular dynamics simulations, which reveal partial folding of the unstructured domains upon complex formation. The authors resort to a multitude of data sources like relaxation data, Ca chemical shifts etc. and combine the findings to a plausible model. Further, they examine the influence of phosphorylation on the affinity of the interaction and identify pT55 as a phosphorylation site that enhances the affinity. On top of characterizing the natural interaction, they also determine details of binding of the potentially therapeutic peptide FOXO4-DRI, which is important for developing peptide-based drugs against cellular senescence.

Overall, the level of characterization of this difficult system is impressive. The paper is well written and understandable, and the results and conclusions are sound, although I had hoped for one or the other additional experiment.

We thank the reviewer for positive evaluation and her/his suggestions and conducted the additional analyses/experiments requested.

The authors for example state that the interaction of p53-TAD2 and FOXO4-FH is mediated by electrostatic and hydrophobic interactions. Was the salt dependence of the affinity measured? Or could key hydrophobic residues be mutated? This would add experimental evidence for this statement. Otherwise, I would rephrase the text that there are strong indications for hydrophobic and electrostatic nature of the interaction.

We conducted additional ITC experiments (Fig S5, Table 1) showing that an increase of the salt concentration from 50 to 300 mM NaCl as well as hydrophobic TAD2 mutations in p53¹⁻⁹⁴ (I50A, W53A, F54A and the corresponding triple mutant) impair p53¹⁻⁹⁴ binding to the FOXO4^{FH}, with non-detectable binding at 300 mM NaCl and a decrease in affinity by a factor 2-9 for the tested point mutations. We included in the main text the following statement:

"In agreement, an increase of the salt concentration or alanine mutations of hydrophobic residues in TAD2 impair binding of p53¹⁻⁹⁴ to FOXO4^{FH} (Fig. S6, Table 1)."

Figure S6. p53¹⁻⁹⁴ interacts with FOXO4^{FH} via a network of hydrophobic and charge-based interactions: Titration of 25 μM of p53¹⁻⁹⁴, p53^{1-94 I50A}, p53^{1-94 W53A}, p53^{1-94 F54A}, p53^{1-94 3A} and p53^{1-94 high salt} with 250 μM of FOXO4^{FH} (a-f). Titration of 50 μM of p53¹⁻⁹⁴ with 500 μM of FOXO4^{DRI} in (g). The reported errors correspond to the SD of the fit.

In the same paragraph, it is stated that DNA and p53-TAD2 have overlapping binding surfaces on FOXO. Would it be a large effort to check if there is actual competition in an experiment?

We now performed NMR experiments showing that addition of a FOXO4 target DNA sequence (CTDSP2) can efficiently compete out p53¹⁻⁹⁴ from FOXO4^{FH}. We could see that specific NMR signals for TAD2 residues could return from the FOXO4^{FH} bound state to the free state upon DNA addition (Fig S2d). We included in the main text the following statement:

“Accordingly, addition of a FOXO4 target DNA sequence (CTDSP2) can efficiently compete out FOXO4^{FH} from p53¹⁻⁹⁴ (Fig. S2d).”

Figure S2. Characterization of FOXO4 - p53 interactions: (a) $^1\text{H},^{15}\text{N}$ HSQC spectrum of $100\ \mu\text{M}$ ^{15}N -labeled FOXO4^{Cter} in the absence (black) or in presence of one stoichiometric equivalent of unlabeled p53¹⁻⁹⁴ (blue). (b) $^1\text{H},^{15}\text{N}$ HSQC spectrum of $100\ \mu\text{M}$ ^{15}N -labeled FOXO4^{Nter} in the absence (black) or presence of one stoichiometric equivalent of unlabeled p53¹⁻⁹⁴ (blue). (c) Overlay of $50\ \mu\text{M}$ ^{15}N -labeled FOXO4^{CR3} in the absence (black) or presence of $200\ \mu\text{M}$ p53^{DBD} (blue) or $50\ \mu\text{M}$ CDKN1A-p53-RE and $200\ \mu\text{M}$ p53^{DBD} (cyan). (d) Overlay of $50\ \mu\text{M}$ ^{15}N -labeled p53¹⁻⁹⁴ in the absence (black) or presence of $100\ \mu\text{M}$ FOXO4^{FH} (orange) or $50\ \mu\text{M}$ CTDSP2 and $100\ \mu\text{M}$ FOXO4^{FH} (cyan).

Also, the authors state that the FOXO4-DRI peptide disrupts the interaction between FOXO4-FH and p53-TAD based on chemical shift perturbations. Could a competition experiment be performed that shows that FOXO4-FH indeed becomes free in presence of FOXO4-DRI? This would dismiss potential doubts about allosteric effects, that CSPs generally can suffer from. While I understand that DNA binding to FOXO4-FH is less in the focus of this manuscript, the competition of FOXO4-DRI and FOXO-FH definitely is, and therefore I would think that this experiment is important for this paper, especially given that the required resources all seem to be present.

We performed an experiment where we titrated FOXO4-DRI to a preformed complex of ^{15}N -labeled FOXO4^{FH} and p53^{TAD2} and could show that FOXO4-DRI can efficiently compete out p53^{TAD2} from FOXO4^{FH} (Fig S7a). This is in line with previous data from our lab showing that FOXO4-DRI can compete p53¹⁻³¹² from FOXO4^{FH} (Baar *et al.* Targeted Apoptosis of Senescent Cells Restores Tissue Homeostasis in Response to Chemotoxicity and Aging. *Cell*. 2017 Mar 23;169(1):132-147.e16. doi: 10.1016/j.cell.2017.02.031. PMID: 28340339). We included in the main text the following statement:

"In agreement, addition of FOXO4-DRI can efficiently compete out p53^{TAD2} from FOXO4^{FH}, as indicated by the chemical shift of $^1\text{H},^{15}\text{N}$ FOXO4^{FH} cross-peaks that return from a p53 bound state to a p53 free state upon addition of FOXO4-DRI (Fig. S7a)."

Figure S7. Characterization of FOXO4-DRI binding. (a) Overlay of 100 μM ^{15}N -labeled FOXO4^{FH} in the absence (black) or presence of 200 μM p53^{TAD2} (blue) or 200 μM FOXO4-DRI and 200 μM p53^{TAD2} (cyan). (b) Overlay of 100 μM ^{15}N -labeled p53^{DBD} in the absence (black) or presence of 200 μM FOXO4-DRI (orange). (c) Pull down of endogenous p53, p53 ^{Δ TAD2} and p53^{KO} with biotinylated biotFOXO4^{DRI} in RPE-1 cells. (d) Plot of the $^{13}\text{C}\alpha/^{13}\text{C}\beta$ secondary chemical shifts of p53¹⁻⁹⁴ in the absence (black) or presence of 400 μM unlabeled FOXO4-DRI (orange). (e) $^{15}\text{N}\{^1\text{H}\}$ NOEs of 300 μM p53¹⁻⁹⁴ in the absence (black) or presence of 400 μM unlabeled FOXO4-DRI (orange). R_1 (f) and R_2 (g) relaxation rates of 300 μM p53¹⁻⁹⁴ in the absence (black) or presence of 400 μM unlabeled FOXO4-DRI (orange).

The data on phosphorylation are nice and convincing. Can the authors speculate on the influence of pSer46 on the alpha helicity of the subsequent residues?

According to the MD analyses, phosphorylation of Ser46 leads to an increase of helicity in Asp48 and Asp49 in the FOXO4^{FH}-bound state when compared to the unphosphorylated version (Figure S12). Interestingly, in the free state, MD analyses showed that phosphorylation of either Ser46, Thr55 or both lead to a loss of helicity of p53^{TAD2} (Figure S10). This indicates that in the FOXO4^{FH} and FOXO4-DRI bound states, helicity is maintained via interactions with FOXO4^{FH} and FOXO4-DRI, respectively. We have updated the text accordingly:

“Importantly, when both residues (pS46 and pT55) are phosphorylated, p53^{TAD2} loses helicity in the absence of binding partners. In contrast, p53^{TAD2} maintains a α -helical conformation when bound to either FOXO4^{FH} or FOXO4-DRI.”

Figure S10. Ramachandran plot for residues 47-54 in unphosphorylated and phosphorylated p53^{TAD2}, respectively along the different MD simulations.

The affinity of FOXO4-DRI to p53(1-94) is stated to be 0.4 μM . (5-fold “higher” than FOXO4-FH. Is that stronger or weaker? I was under the impression that DRI binds more strongly.) However, all later ITC measurements indicate 10–100 fold weaker affinity, and NMR spectra appear to be in fast exchange, which would not support a nanomolar K_d . This seeming disparity should be discussed in the paper.

We agree with the referee that the word “higher” is a wrong statement here and changed in the main text this word by “stronger”. The later affinities are towards binding to p53^{TAD2} and not p53¹⁻⁹⁴ and indeed weaker, thus indicating a contribution of p53^{TAD1} to FOXO4^{FH}/FOXO4-DRI binding. Nevertheless, we could show that p53^{TAD1} on its own is insufficient to interact with FOXO4, indicating that p53^{TAD2} is the main binding site.

The differences between the changes of NMR signals (fast exchange) and the ITC data could in part be due the different buffers used for these experiments. NMR titration experiments have been carried out in 50 mM NaH₂PO₄/Na₂HPO₄, pH 7.5, whereas ITC has been carried out in 20 mM Hepes pH 7.0, 50 mM NaCl, 2 mM tris(2-carboxyethyl)phosphine (TCEP). The difference in buffer concentration, pH and potential interference of phosphate with the interaction might explain the differences.

The structure reveals several insights into the interaction. Here I have a more general remark: is it the right approach to chase down the ONE lowest energy conformation of such a complex? It reminds me of the controversy around chaperone-client interactions, where it was tried to find the preferred binding mode of a client protein, although the better representation is certainly several different binding modes of similar binding energy. I suspect that this interaction might also have this characteristic, in the sense that there are several conformations that are close in binding energy. The presented conformation was “artificially” enriched by calculating 1000 conformers instead of the 100 usual, hydrogen bonds were enforced, and the MD started only from the lowest energy structure. Is there a way to represent this complex with several favorable conformations? I suspect their relative binding energies are not too far apart from each other. Maybe the authors can discuss whether their data rather indicates that there is a single conformation of the complex as “underlying truth” and that the low density/low accuracy of restraints lead to a somewhat fuzzy model, or that the determined structure is one preferred conformation among several interaction modes that have similar energy.

We agree that representing the complex structures with one lowest energy conformation does not provide insights into the dynamics of the system and other states that might be sampled in these dynamic complexes. Therefore, and to avoid over-restraining the system, we carried out restraint-free MD simulations of the complexes and used the structure of the lowest energy state to prepare the figures. We also showed the RMSD/RMSF of the MD calculations to outline the flexibility.

To better address the comments of the reviewer and to visualize the dynamics of the system, we analyzed the conformational space of p53^{TAD2} alone and in complex with the two ligands FOXO4^{FH} and FOXO4-DRI using Markov state models [<https://doi.org/10.1006/jcph.1999.6231>; <https://doi.org/10.1063/1.1738647>; <https://doi.org/10.1063/1.2714538>; <https://doi.org/10.1021/jp0761665>]. We selected the backbone torsions as features and the original space was reduced using time-structure independent component analysis (tICA). [<https://doi.org/10.1063/1.4811489>; <https://doi.org/10.1063/1.4811489>; <https://doi.org/10.1021/acs.jctc.5b00553>] After choosing the number of dimensions, the conformational space sampled in the MD simulations was divided into microstates using the clustering algorithm k-means. The models were validated using the Chapman-Kolmogorov test and the resulting free energy surfaces and transition probabilities were analyzed. We analyzed the three systems and calculated both stationary distribution and its related free energy for each model (Figures S5 and S8).

As expected, several minima were found in all systems. Interestingly, we observed that p53^{TAD2} binding to FOXO4^{FH} or FOXO4-DRI deeply impact the distribution of microstates into the two time-lagged independent components (ICs). Interestingly, binding events to FOXO4^{FH} or to FOXO4-DRI reduced dispersion in IC1 vs IC2. In all complexes the conformational changes are restricted, indicating increased rigidity of the structures. These results and the Ramachandran plots, confirm that p53^{TAD2}

gets a significant secondary structure component in most of complexes. Indeed, when taking together non-bound and bound states, the low-energy regions in the reweighted free energy surfaces correspond to latter ones.

We updated the text and the methods accordingly:

“To better understand the dynamics of p53^{TAD2} when bound to FOXO4^{FH}, we used Markov state models to analyze the conformational space of p53^{TAD2} both in isolation and in complex with FOXO4^{FH}. As anticipated, several minima were identified, revealing that the binding of p53^{TAD2} to FOXO4^{FH} alters the distribution of microstates within the two time-lagged independent components (ICs; Fig. S5). Notably, binding to FOXO4^{FH} reduces dispersion between IC1 and IC2, resulting in more restricted conformational changes. This suggests that p53^{TAD2} becomes increasingly rigid upon interaction with FOXO4^{FH}.”

“Markov state modeling of p53^{TAD2} both in isolation and in complex with FOXO4-DRI displayed several minima, revealing that the binding of p53^{TAD2} to FOXO4-DRI alters the distribution of microstates within the two time-lagged independent components (Figure S8). As in case of FOXO4^{FH} binding, binding to FOXO4-DRI reduces dispersion between IC1 and IC2, resulting in more restricted conformational changes and increasing rigidity in the bound state.”

Figure S5. Markov State Analysis of the p53^{TAD2}-FOXO4^{FH} complex. (a) Distribution of p53^{TAD2} microstates projected onto the first two time-lagged independent components (ICs) at lag time t=5 ns. The distribution of microstates was grouped and modeled into three distinct macrostates (S₁-S₃, right). (b) Stationary distribution and reweighted free energy surface of p53^{TAD2} projected onto the first two time-lagged independent components (ICs) at lag time t=5 ns. (c) Ensemble of representative structures of p53^{TAD2} (orange) bound to FOXO4^{FH} (dark blue) in the three macrostates displayed in (a).

Figure S8. Markov State Analysis of the p53^{TAD2}-FOXO4-DRI complex. (a) Distribution of p53^{TAD2} microstates projected onto the first two time-lagged independent components (ICs) at lag time $t=5$ ns. The distribution of microstates was grouped and modeled into five distinct macrostates (S_1 - S_5 , right). (b) Stationary distribution and reweighted free energy surface of p53^{TAD2} projected onto the first two time-lagged independent components (ICs) at lag time $t=5$ ns. (c) Ensemble of representative structures of p53^{TAD2} (orange) bound to FOXO4-DRI (dark blue) in the three macrostates displayed in (a).

The manuscript should only be accepted upon reception of the BMRB codes. However, I'm very happy to see the submission files and the assignments in user-friendly Xeas format.

The assignments of FOXO4^{FH}, FOXO4^{CR3}, p53^{TAD2} and p53¹⁻⁹⁴ had already been submitted to the BMRB under the accession numbers 50398, 50403, 51125 and 51124, respectively. Resonance assignments of FOXO4-LRI and p53^{TAD1} were submitted to the BRMB during the reviewing process under the accession numbers 52457 and 52458, respectively. We have updated the method part accordingly.

There are a number of typos in the manuscript. Here are a few:

“Aging” and “ageing” are both used, e.g. lines 36 and 38. Stick to one.

Line 66: “coping” instead of copying

Line 140: there might be a mix up between labeled and unlabeled proteins.

Line 301: comma missing after “summarizing”

Line 427: were affected “more strongly”

Line 438: “we” conducted...

Line 513: “efforts” instead of affords

According to the reviewer's comment, we corrected all typos in the main text.

Overall, I'd like to congratulate the authors for a nice piece of work which certainly deepens our insights into the workings of FOXO4 and p53, and showing potential avenues for future treatment of senescence.

We thank the reviewer for her/his thoughtful comments.

Reviewer #2 (Remarks to the Author):

In the ms, the authors mainly present the solution NMR structure complex of p53 with FOXO4 (forkhead or FH domain) or FOXO4-DRI, with FOXO4 the Forkhead box protein O 4 or a transcription factor (TF) of FOX family, and FOXO4-DRI a disordered cell-penetrating peptide as a drug molecule to target on p53 interaction.

Although the authors provided fairly thorough background information about the studied systems, some basic questions were not explained nor brought about to attention. For example, what is the molecular role of the FOXO4-p53 interaction, e.g., in regard to FOXO4-DNA and p53-DNA interactions or regulation functions, respectively? Since FOXO4 is suggested to particularly interact with p53-TAD2, while TAD2 is supposed to interact with the DBD (DBD of p53 or DBD of FOXO4 or other TFs? on page 3 line 85), can one expect that FOXO4 interferes with p53 promoter recognition? Any further literature evidences? Similarly, how is the TF function of FOXO4 impacted by its interaction with p53? Btw, is the FH domain also the DBD of FOXO4?

We thank the reviewer for her/his thoughtful comments. It is yet not demonstrated whether p53 and FOXO4 can synergistically contribute to the activation/inhibition of the transcription of targeted genes. We show that the binding surface of p53 and FOXO4 TADs on the DNA binding domain of FOXO4 (FH domain) and p53 (DBD) overlap with the DNA binding interface. We now have included an experiment showing that indeed addition of DNA can compete out p53 binding to FOXO4^{FH} (Fig. S2c) and FOXO4 binding to p53^{DBD} (Fig. S2d), suggesting that p53 and FOXO4 could regulate each other in terms of transcription activation of specific gene target.

Figure S2. Characterization of FOXO4 - p53 interactions: (a) ¹H, ¹⁵N HSQC spectrum of 100 μM ¹⁵N-labeled FOXO4^{Cter} in the absence (black) or in presence of one stoichiometric equivalent of unlabeled p53¹⁻⁹⁴ (blue). (b) ¹H, ¹⁵N HSQC spectrum of 100 μM ¹⁵N-labeled FOXO4^{Nter} in the absence (black) or presence of one stoichiometric equivalent of unlabeled p53¹⁻⁹⁴ (blue). (c) Overlay of 50 μM ¹⁵N-labeled FOXO4^{CR3} in the absence (black) or presence of 200 μM p53^{DBD} (blue) or 50 μM CDKN1A-p53-RE and 200 μM p53^{DBD} (cyan). (d) Overlay of 50 μM ¹⁵N-labeled p53¹⁻⁹⁴ in the absence (black) or presence of 100 μM FOXO4^{FH} (orange) or 50 μM CTDSP2 and 100 μM FOXO4^{FH} (cyan).

To the best of our knowledge, there are no comprehensive studies related to the co-occurrence of p53 and FOXO binding sites within promoter regions and this remains to be studied. The best known promoter region regulated by p53 and FOXO is the promoter region of the cell cycle regulator p21. *CDKN1A*, the gene encoding p21^{Cip1}, shows a canonical FOXO target sequence to be flanked by two p53 binding sites (Baar *et al.* Targeted Apoptosis of Senescent Cells Restores Tissue Homeostasis in Response to Chemotoxicity and Aging. *Cell*. 2017 Mar 23;169(1):132-147.e16. doi: 10.1016/j.cell.2017.02.031. PMID: 28340339). The p53 promoter region contains binding sites for the FOX proteins FOXL2 and FOXA1 (Choi Y *et al.* FOXL2 and FOXA1 cooperatively assemble on the TP53 promoter in alternative dimer configurations. *Nucleic Acids Res*. 2022 Aug 26;50(15):8929-8946. doi: 10.1093/nar/gkac673. PMID: 35920317), but whether FOXOs can bind the p53 promoter region has not been studied so far.

L

(L) Schematic representation of the p21^{Cip1} (*CDKN1a*) promoter in which the canonical FOXO target sequence is flanked by two p53 binding sites (taken from Baar *et al.*).

With respect to the impact of the p53-FOXO4 interaction on the DNA-binding affinities, Mandal *et al.* (Mandal R *et al.* FOXO4 interacts with p53 TAD and CRD and inhibits its binding to DNA. *Protein Sci*. 2022 May;31(5):e4287. doi: 10.1002/pro.4287. PMID: 35481640) found that complex formation reduces the DNA-binding affinity of p53 but not of FOXO4. However, the consequences of p53/FOXO4 transcriptional activity remain elusive.

Beyond that, transient interactions involving multiple binding sites might favor the formation of p53-FOXO4 transcriptional condensates and in turn favor both p53 and FOXO4-dependent transcription. Previous studies demonstrated that *in vitro* recombinant p53 can phase separate (Kamagata K *et al.* Liquid-like droplet formation by tumor suppressor p53 induced by multivalent electrostatic interactions between two disordered domains. *Sci Rep*. 2020 Jan 17;10(1):580. doi: 10.1038/s41598-020-57521-w. PMID: 31953488). We also have first evidences of p53-FOXO4 co-partitioning in condensates, but we think that this goes beyond the scope of this manuscript.

We now discuss the potential role of p53-FOXO4 interaction in gene regulation and/or in the formation of biomolecular condensates in the discussion section:

*“It remains to be demonstrated whether p53 and FOXO4 can synergistically regulate targeted gene transcription. Our findings indicate that the binding surface of p53/FOXO4 TADs on their DNA binding domains overlap with their respective DNA binding interface. Notably, the addition of DNA competes with p53 binding to FOXO4^{FH} (Fig. S2d) and FOXO4 binding to p53^{DBD} (Fig. S2c). This suggests that p53 and FOXO4 may form complexes within promoter regions. Furthermore, given that the p53/FOXO4 TADs are involved in self-interaction via binding to their own DNA binding domains,^{14, 41} another regulatory mechanism could involve the formation of biomolecular condensates. Evidence supports this possibility, as recombinant p53 has been shown to phase separate *in vitro*.⁴⁵ It is plausible that transient interactions between multiple binding sites, including p53-p53, FOXO4-FOXO4 and p53-FOXO4, facilitate the formation of p53-FOXO4 condensates. These condensates may, in turn, play an important role in p53/FOXO4-mediated transcription, as well as other processes like PML body recruitment or the DNA-damage response. Supporting this idea, we previously observed that in senescent cells, FOXO4 and p53 localize within PML bodies and DNA-SCARS. Interestingly, the addition of FOXO4-DRI disrupts this localization, leading to the exclusion of p53 from these condensates.^{5”}*

The authors later show that “The interface on FOXO4FH covered by p53TAD2 overlaps with the one bound to DNA indicating a competition between the binding events”. Are there additional experimental or literature evidences supporting that DNA and p53-TAD2 compete for the FOXO-FH binding?

We now performed NMR experiments showing that addition of a FOXO4 target DNA sequence (CTDSP2) can efficiently compete out p53¹⁻⁹⁴ from FOXO4^{FH}. We could see that specific NMR signals for TAD2 residues could return from the FOXO4 FH bound state to the free state upon DNA addition (Fig S2c). We included in the main text the following statement:

“Accordingly, addition of a FOXO4 target DNA sequence (CTDSP2) can efficiently compete out FOXO4^{FH} from p53¹⁻⁹⁴ (Fig. S2d).”

On page 8-9, the authors address “the FOXO4FH-p53TAD2 complex structure reveals overlapping binding surfaces with DNA”, “Fig 3a shows an overlay of the top 10 lowest-energy models”, and with extensive MD simulations” conducted. How were the energy models generated? The authors should provide full technical descriptions and procedures to justify the modeling part, and provide structural models and files for future validation. Similarly, how about the FOXO4-DRI-p53 interaction models?

We have further extended the description of the model generation and the MD calculations. The structural models and MD data have been deposited to Zenodo (<https://zenodo.org/doi/10.5281/zenodo.10963887>)

For the MD simulations, it was not clear how extensive or how long (and how many) the simulations were conducted, whether the systems were well equilibrated (e.g., by showing RMSDs overall simulation time), or the interaction interfaces were stabilized, or say, the results shown in Fig 3 converged or not. Similar concerns apply also to the FOXO4-DRI-p53 simulations.

We have further extended the description of MD simulations.

Reviewer #3 (Remarks to the Author):

The manuscript “The disordered p53 transactivation domain is the target of FOXO4 and the senolytic compound FOXO4-DRI” by Bourgeois et al. describes a structural and biophysical characterization of the interaction between the p53 N-terminal region 1-94 (including both TAD domains) and the Forkhead domain of FOXO4 (FOXO4-FH). This manuscript is a structural follow-up on a seminal paper from 2016, where the authors could show that perturbing the FOXO4-p53 interaction induces apoptosis of senescent cells. The current manuscript aims to provide a structural basis of this phenomenon using purely *in vitro*-based biophysics and structural biology. The regions involved in this interaction were determined by NMR chemical shift perturbations (CSPs) and a structural model of FOXO4-FH with p53-TAD2 was provided based on NOEs, dihedral angles, CSPs, paramagnetic relaxation enhancements (PREs) with additional support from MD simulations. Similarly, they could demonstrate how the FOXO4-DRI peptide discovered earlier binds to p53-TAD2 by NMR spectroscopy. This interaction would compete with FOXO4 for p53 binding and has an effect on senescent cells.

While this is an impactful interaction and detailed knowledge on this is important, the study misses a thorough validation of the structural findings in form of a mutational analysis. This is surprising, as the authors seem to have the means to perform these experiments. This would be done preferably *in vivo* or at least in cellular assays, if possible, to show that the identified residues are indeed important. This is especially important, as the interaction is based on rather unspecific charge-charge interactions, which altogether seem to be rather weak, as all NMR titration experiments exhibit weak CSPs in the fast exchange regime (Figure 1 and related Supplementary Figures). This is the main major concern I have with this study.

We thank the reviewer for her/his thoughtful comments. We have tested the contribution of p53^{TAD2} in FOXO4^{FH} and FOXO4-DRI binding in a “semi-cellular” assay with the capabilities we have in our lab. We show that deletion of TAD2 lead to a decrease of binding when using either *in vitro* biotinylated FOXO^{FH} or FOXO-DRI as bait (Fig. S1d and Fig. S7c). In addition, we now have included ITC experiments showing that alanine mutation of key hydrophobic TAD2 residues (i.e. I50, W53 and F54) impair binding to FOXO4^{FH} (Fig S6, Table 1).

Figure S1: p53^{TAD2} is the primary binding site of FOXO4^{FH}. (a) Overlay of ¹H,¹⁵N HSQC spectra of ¹⁵N-labeled p53^{DBD} at 100 μM in the absence (black) and presence (orange) of 100 μM FOXO4^{FH}. (b) Overlay of ¹H,¹⁵N HSQC spectra of ¹⁵N-labeled p53^{TAD1} at 100 μM in absence (black) and presence (orange) of 100 μM FOXO4^{FH}. (c) Overlay of ¹H,¹⁵N HSQC spectra of ¹⁵N-labeled FOXO4^{FH} at 100 μM in absence (black) or presence (orange) of 100 μM p53^{TAD1}. (d) Pull down of endogenous p53, p53^{ΔTAD2} and p53^{KO} with biotinylated biotFOXO4^{FH} in RPE-1 cells.

Figure S7. Characterization of FOXO4-DRI binding. (a) Overlay of 100 μM ^{15}N -labeled FOXO4^{FH} in the absence (black) or presence of 200 μM p53^{TAD2} (blue) or 200 μM FOXO4-DRI and 200 μM p53^{TAD2} (cyan). (b) Overlay of 100 μM ^{15}N -labeled p53^{DBD} in the absence (black) or presence of 200 μM FOXO4-DRI (orange). (c) Pull down of endogenous p53, p53 ^{Δ TAD2} and p53^{KO} with biotinylated biotFOXO4^{DRI} in RPE-1 cells. (d) Plot of the $^{13}\text{C}\alpha$ / $^{13}\text{C}\beta$ secondary chemical shifts of p53¹⁻⁹⁴ in the absence (black) or presence of 400 μM unlabeled FOXO4-DRI (orange). (e) $^{15}\text{N}\{^1\text{H}\}$ NOEs of 300 μM p53¹⁻⁹⁴ in the absence (black) or presence of 400 μM unlabeled FOXO4-DRI (orange). R_1 (f) and R_2 (g) relaxation rates of 300 μM p53¹⁻⁹⁴ in the absence (black) or presence of 400 μM unlabeled FOXO4-DRI (orange).

Figure S6. p53¹⁻⁹⁴ interacts with FOXO4^{FH} via a network of hydrophobic and charge-based interactions: Titration of 25 μM of p53¹⁻⁹⁴, p53¹⁻⁹⁴ I50A, p53¹⁻⁹⁴ W53A, p53¹⁻⁹⁴ F54A, p53¹⁻⁹⁴ 3A and p53¹⁻⁹⁴ high salt with 250 μM of FOXO4^{FH} (a-f). Titration of 50 μM of p53¹⁻⁹⁴ with 500 μM of FOXO4^{DRI} in (g). The reported errors correspond to the SD of the fit.

Although an importance of the FOXO4-p53 has been shown and this seems to be the only regions identified, the interaction seems so weak that it is hard to fathom how this will compete with all the other proteins and DNA, p53 and FOXO4 interact with. The same region of FOXO4 interacts with DNA, which is probably much stronger than with p53-TAD2 (hard to say, as the authors do not provide affinities for their FOXO4-p53-TAD2 interaction, although the data for that seems to be present in the NMR titrations). Also, the FOXO4-CR3 region interacts at the same spot with its own FOXO4-FH domain, which is covalently attached and therefore not diffusion-limited.

The interaction of p53 and FOXO4 involves multiple binding sites, (i) p53^{TAD} binding to the FOXO4^{FH} (ii) FOXO4^{TAD} binding to the p53^{DBD} (iii) FOXO4 N-terminal region binding to the p53^{DBD} that could synergistically contribute to an enhanced affinity in the context of the full length proteins. We have included additional data showing that recombinant biotinylated FOXO4^{FH} can pull-down endogenous p53 in cells and that the TAD2 region is important for this interaction (Fig S1d, see above). Furthermore, we included additional data showing that the binding surface of p53 and FOXO4 TADs overlap with the DNA binding site of FOXO4 (FH domain) and p53 (DBD) by showing that addition of DNA can compete out p53 binding to FOXO4^{FH} (Fig. S2d) and FOXO4 binding to p53^{DBD} (Fig. S2c), suggesting that p53 and FOXO4 could regulate each other. Given that the p53 and FOXO4 TADs are involved in self-interaction via binding to their own DNA binding domain, one other possibility of regulatory function is the formation of biomolecular condensates as mentioned by the reviewer.

Figure S2. Characterization of FOXO4 - p53 interactions: (a) $^1\text{H},^{15}\text{N}$ HSQC spectrum of $100\ \mu\text{M}$ ^{15}N -labeled FOXO4^{Cter} in the absence (black) or in presence of one stoichiometric equivalent of unlabeled p53¹⁻⁹⁴ (blue). (b) $^1\text{H},^{15}\text{N}$ HSQC spectrum of $100\ \mu\text{M}$ ^{15}N -labeled FOXO4^{Nter} in the absence (black) or presence of one stoichiometric equivalent of unlabeled p53¹⁻⁹⁴ (blue). (c) Overlay of $50\ \mu\text{M}$ ^{15}N -labeled FOXO4^{CR3} in the absence (black) or presence of $200\ \mu\text{M}$ p53^{DBD} (blue) or $50\ \mu\text{M}$ CDKN1A-p53-RE and $200\ \mu\text{M}$ p53^{DBD} (cyan). (d) Overlay of $50\ \mu\text{M}$ ^{15}N -labeled p53¹⁻⁹⁴ in the absence (black) or presence of $100\ \mu\text{M}$ FOXO4^{FH} (orange) or $50\ \mu\text{M}$ CTDSP2 and $100\ \mu\text{M}$ FOXO4^{FH} (cyan).

To the best of our knowledge, there are no comprehensive studies related to the co-occurrence of p53 and FOXO binding sites within promoter regions and this remains to be studied. The best known promoter region regulated by p53 and FOXO is the promoter region of the cell cycle regulator p21. *CDKN1A*, the gene encoding p21^{Cip1}, shows a canonical FOXO target sequence to be flanked by two p53 binding sites (Baar *et al.* Targeted Apoptosis of Senescent Cells Restores Tissue Homeostasis in Response to Chemotoxicity and Aging. *Cell*. 2017 Mar 23;169(1):132-147.e16. doi: 10.1016/j.cell.2017.02.031. PMID: 28340339). The p53 promoter region contains binding sites for the FOX proteins FOXL2 and FOXA1 (Choi Y *et al.* FOXL2 and FOXA1 cooperatively assemble on the TP53 promoter in alternative dimer configurations. *Nucleic Acids Res*. 2022 Aug 26;50(15):8929-8946. doi: 10.1093/nar/gkac673. PMID: 35920317), but whether FOXOs can bind the p53 promoter region has not been studied so far.

L

(L) Schematic representation of the p21Cip1 (CDKN1a) promoter in which the canonical FOXO target sequence is flanked by two p53 binding sites (taken from Baar *et al.*).

With respect to the impact of the p53-FOXO4 interaction on the DNA-binding affinities, Mandal *et al.* (Mandal R *et al.* FOXO4 interacts with p53 TAD and CRD and inhibits its binding to DNA. *Protein Sci.* 2022 May;31(5):e4287. doi: 10.1002/pro.4287. PMID: 35481640) found that complex formation reduces the DNA-binding affinity of p53 but not of FOXO4. However, the consequences of p53/FOXO4 transcriptional activity remain elusive.

We now discuss the potential role of p53-FOXO4 interaction in gene regulation and/or in the formation of biomolecular condensates in the discussion section:

*"It remains to be demonstrated whether p53 and FOXO4 can synergistically regulate targeted gene transcription. Our findings indicate that the binding surface of p53/FOXO4 TADs on their DNA binding domains overlap with their respective DNA binding interface. Notably, the addition of DNA competes with p53 binding to FOXO4^{FH} (Fig. S2d) and FOXO4 binding to p53^{DBD} (Fig. S2c). This suggests that p53 and FOXO4 may form complexes within promoter regions. Furthermore, given that the p53/FOXO4 TADs are involved in self-interaction via binding to their own DNA binding domains,^{14, 41} another regulatory mechanism could involve the formation of biomolecular condensates. Evidence supports this possibility, as recombinant p53 has been shown to phase separate *in vitro*.⁴⁵ It is plausible that transient interactions between multiple binding sites, including p53-p53, FOXO4-FOXO4 and p53-FOXO4, facilitate the formation of p53-FOXO4 condensates. These condensates may, in turn, play an important role in p53/FOXO4-mediated transcription, as well as other processes like PML body recruitment or the DNA-damage response. Supporting this idea, we previously observed that in senescent cells, FOXO4 and p53 localize within PML bodies and DNA-SCARS. Interestingly, the addition of FOXO4-DRI disrupts this localization, leading to the exclusion of p53 from these condensates.^{5"}*

One possibility where this weak interaction could be of relevance is in biomolecular condensates. This interaction seems to occur in PML bodies, which, to my knowledge are biomolecular condensates. However, the authors do not consider this in their introduction or discussion, although it could be a neat explanation. There are reports about p53 condensation properties. Nevertheless, a mutational analysis is needed to provide a validation of their structure.

Are the identified residues known as disease-related mutations? In the meanwhile, there should be enough cancer omics data that even rare disease-mutations can be compared.

We included a mutational analysis using ITC and show that alanine mutation of key hydrophobic TAD2 residues (ie. I50, W53 and F54) impair binding to FOXO4^{FH} (Fig S6, Table 1).

We agree with reviewer that one can speculate that such transient interaction involving multiple binding sites might favor the formation of p53-FOXO4 transcriptional condensates and or PML body formation and in turn favor both p53 and FOXO4-dependent transcription and/or DNA-damage response. In this line, we previously observed that in senescent cells FOXO4 and p53 co-localize in PML bodies / DNA-SCARS and that addition of FOXO4-DRI leads to p53 exclusion from these bodies (Baar *et al.* Targeted Apoptosis of Senescent Cells Restores Tissue Homeostasis in Response to Chemotoxicity and Aging. *Cell.* 2017 Mar 23;169(1):132-147.e16. doi: 10.1016/j.cell.2017.02.031. PMID: 28340339). Previous studies demonstrated that *in vitro* recombinant p53 can phase separate (Kamagata K *et al.* Liquid-like droplet formation by tumor suppressor p53 induced by multivalent electrostatic interactions between two disordered domains. *Sci Rep.* 2020 Jan 17;10(1):580. doi: 10.1038/s41598-020-57521-w. PMID: 31953488). We also have first evidences of p53-FOXO4 co-partitioning in condensates, but we think that this goes beyond the scope of this manuscript.

We now discuss the potential role of p53-FOXO4 interaction in the formation of biomolecular condensates in the discussion section (see above).

Figure S6. $p53^{1-94}$ interacts with $FOXO4^{FH}$ via a network of hydrophobic and charge-based interactions: Titration of 25 μM of $p53^{1-94}$, $p53^{1-94 I50A}$, $p53^{1-94 W53A}$, $p53^{1-94 F54A}$, $p53^{1-94 3A}$ and $p53^{1-94 \text{high salt}}$ with 250 μM of $FOXO4^{FH}$ (a-f). Titration of 50 μM of $p53^{1-94}$ with 500 μM of $FOXO4^{DRI}$ in (g). The reported errors correspond to the SD of the fit.

(Left) Structured illumination microscopic (SIM) image of the nucleus of a senescent IMR90 cell stained for FOXO4, 53BP1, and PML. Yellow arrow: Area processed for 3D surface-rendering (insets). (Right) FOXO4 and Ser15-phosphorylated p53 in senescent IMR90. Intensity plot (arbitrary units) of individual pixels measured by the indicated line. (Figure taken from Baar et al.)

Another major concern is the presentation of the NMR data. I cannot find any statistics table for the NMR structures. The structures are not submitted to the PDB and to my knowledge this is required by the journal. It is difficult with structures based on integrative modelling, but here a classical NMR approach is used supplemented with PREs. The authors claim to have included 48 intermolecular NOEs in the structure calculation, but they are nowhere shown also not in the provided BMRB files (they include only the chemical shift assignments). For such a small interaction surface, 48 intermolecular NOEs are a lot, and they could have been already enough for obtaining an NMR structure ensemble? Some exemplary NOE strips in the supplement would really strengthen the manuscript. In the results part, it is somewhat mentioned in passing that due to intermediate exchange and extensive line broadening only a few NOE-based distance restraints were obtained. Again, 48 intermolecular NOEs are not a few. At least in the methods section it says 48 were included in the structure calculation.

Given that the p53^{TAD2}-FOXO4^{FH} and p53^{TAD2}-FOXO4-DRI complexes remain dynamic (see MD), we refrained from carrying out such an analysis and deposition to the PDB as done for conventional NMR structures (see also comment by reviewer 1). To address the challenge of analyzing and representing the dynamic complexes, we carried out extensive MD runs and analyses which we have further extended in the revised version of the manuscript. We deposited the structures and MD analyses to Zenodo (<https://zenodo.org/doi/10.5281/zenodo.10963887>).

We updated the text and the methods accordingly:

“To better understand the dynamics of p53^{TAD2} when bound to FOXO4^{FH}, we used Markov state models to analyze the conformational space of p53^{TAD2} both in isolation and in complex with FOXO4^{FH}. As anticipated, several minima were identified, revealing that the binding of p53^{TAD2} to FOXO4^{FH} alters the distribution of microstates within the two time-lagged independent components (ICs; Fig. S5). Notably, binding to FOXO4^{FH} reduces dispersion between IC1 and IC2, resulting in more restricted conformational changes. This suggests that p53^{TAD2} becomes increasingly rigid upon interaction with FOXO4^{FH}.”

“Markov state modeling of p53^{TAD2} both in isolation and in complex with FOXO4-DRI displayed several minima, revealing that the binding of p53^{TAD2} to FOXO4-DRI alters the distribution of microstates within the two time-lagged independent components (Figure S8). As in case of FOXO4^{FH} binding, binding to FOXO4-DRI reduces dispersion between IC1 and IC2, resulting in more restricted conformational changes and increasing rigidity in the bound state.”

Figure S5. Markov State Analysis of the p53^{TAD2}-FOXO4^{FH} complex. (a) Distribution of p53^{TAD2} microstates projected onto the first two time-lagged independent components (ICs) at lag time t=5 ns. The distribution of microstates was grouped and modeled into three distinct macrostates (S₁-S₃, right). (b) Stationary distribution and reweighted free energy surface of p53^{TAD2} projected onto the first two time-lagged independent components (ICs) at lag time t=5 ns. (c) Ensemble of representative structures of p53^{TAD2} (orange) bound to FOXO4^{FH} (dark blue) in the three macrostates displayed in (a).

Figure S8. Markov State Analysis of the p53^{TAD2}-FOXO4-DRI complex. (a) Distribution of p53^{TAD2} microstates projected onto the first two time-lagged independent components (ICs) at lag time $t=5$ ns. The distribution of microstates was grouped and modeled into five distinct macrostates (S_1 - S_5 , right). (b) Stationary distribution and reweighted free energy surface of p53^{TAD2} projected onto the first two time-lagged independent components (ICs) at lag time $t=5$ ns. (c) Ensemble of representative structures of p53^{TAD2} (orange) bound to FOXO4-DRI (dark blue) in the three macrostates displayed in (a).

We apologize for the confusion raised regarding the restraints and updated the methods part and the supplement accordingly.

We included exemplary NOE strips (Fig S13). It is worth noting that most of the intermolecular NOE-based restraints used in the calculations are ambiguous (i.e. 20/2 non-ambiguous NOEs out of 119/48 total intermolecular NOEs for p53^{TAD2}-FOXO4-DRI/p53^{TAD2}-FOXO4^{FH} complexes, respectively). PRE data were used only for structure determination of the p53^{TAD2}-FOXO4^{FH} complex. We apologize for the confusion and updated the methods part.

Figure S13. Exemplary NOE strips showing intermolecular NOEs between ¹H,¹³C labeled p53^{TAD2} and unlabeled FOXO4^{DRI} extracted from ¹³C,¹⁵N filtered, ¹³C-edited 3D NOESY-HSQC experiments.

Also, the NMR titrations in Figure 1 are clear fast exchange to me, at least from the few titration points shown. From the KDs determined by ITC between p53-1-94 and FOXO4-FH, intermediate line broadening could be expected, but ITC data for FOXO4-FH and p53-TAD2 is missing (it should be provided). There is also only an ITC table and no data shown as opposed to the phosphorylation experiments, where the ITC data is shown in the supplement (but NOT in the table).

The differences between the changes of NMR signals (fast exchange) and the ITC data could in part be due the different buffers used for these experiments. NMR titration experiments have been carried out in 50 mM NaH₂PO₄/Na₂HPO₄, pH 7.5, whereas ITC has been carried out in 20 mM Hepes pH 7.0, 50 mM NaCl, 2 mM tris(2-carboxyethyl)phosphine (TCEP). The difference in buffer concentration, pH and potential interference of phosphate with the interaction might explain the differences.

We added the requested ITC table and data (Fig. S6, Fig. S9, Table 1). We show ITC data related to FOXO4^{FH} and p53^{TAD2} (Fig S9a, Table 1) but due to the very low enthalpy-derived signal even at high protein concentration (50 μM/500 μM) we were not confident in the fit of the data. Nevertheless, it appears that the affinity is also in the micromolar range.

Figure S6. p53¹⁻⁹⁴ interacts with FOXO4^{FH} via a network of hydrophobic and charge-based interactions: Titration of 25 μM of p53¹⁻⁹⁴, p53^{1-94 I50A}, p53^{1-94 W53A}, p53^{1-94 F54A}, p53^{1-94 3A} and p53^{1-94 high salt} with 250 μM of FOXO4^{FH} (a-f). Titration of 50 μM of p53¹⁻⁹⁴ with 500 μM of FOXO4^{DRI} in (g). The reported errors correspond to the SD of the fit.

Figure S9. Phosphorylation of p53^{TAD2} enhances its affinity for FOXO4^{FH} and FOXO4-DRI: Titration of 500 μM of p53^{TAD2}, p53^{TAD2pS46}, p53^{TADpT55}, and p53^{TAD2pS46T55} to 50 μM FOXO4^{FH} (A-D) or FOXO4-DRI (E-H). The reported errors correspond to the SD of the fit.

The authors acquired PRE data. There is quite a bit of confusion with the PREs. While supplementary figure shows four FOXO4-FH variants with engineered spin label sites, in the methods section (structure determination) it says that 5 are included in the calculation. Also, the mutated residues are different from the figure, “structure determination” method section and in the “paramagnetic labelling” section. The PRE plots do not have error bars. At least the authors should show the HSQCs in the para- and diamagnetic state and some zooms to certain residues to estimate how good and reliable the data is.

We agree with the referee and apologize for the confusion and mistakes in the writing/analysis corresponding to the PRE section. We now included in **Figure S4** for each spin-labeled used, the overlay of HSQCs in the para- and diamagnetic state as well as corrected analysis of the PRE plots including errors.

Figure S4. PRE restraints used for the determination of the structure of FOXO4^{FH} – p53^{TAD2}. Paramagnetic relaxation enhancement (PRE) measurements of 100 μM ¹⁵N-labeled FOXO4^{FH} in presence of p53^{TAD2} spin labeled at position 46 or 52 in (a) and (b), respectively. PRE data of 100 μM ¹⁵N-labeled p53^{TAD2} in the presence of FOXO4^{FH} spin labeled at position 112, 140, 145, 159 and 162 in (c) The bar plots show the intensity ratio of the ¹H, ¹⁵N HSQC FOXO4^{FH} in (a, b) or p53^{TAD2} (c) obtained in the paramagnetic divided by the diamagnetic state. Error bars were calculated based on the standard deviation of noise in the paramagnetic and diamagnetic spectra and using error propagation.

To me it seems that the effects are stronger for the p53_1-94 than for the single TADs, as if both TAD bind stronger (multivalency?). However, in the reversed titration this does not seem to be the case.

We agree with the referee that it seems that TAD1 contributes to binding for both FOXO4^{FH} and FOXO4-DRI as indicated by stronger CSPs of ¹⁵N-labeled p53^{TAD2} versus p53¹⁻⁹⁴ cross-peaks upon FOXO4^{FH} addition. However, we could show that p53^{TAD1} on its own is insufficient to interact with FOXO4, indicating that p53^{TAD2} is the main binding site. We included in the main text the following statement:

“We note that the affinity of FOXO4-DRI for p53^{TAD2} is weaker than for p53¹⁻⁹⁴, indicating a contribution of TAD1 in binding”.

The sentence on line 229: However, TAD2 did not exhibit secondary structure elements in its free state but does so in presence of FOXO4-FH. This is a bit of an overstatement. In Figure S3A, I see three black and three orange positive bars with still very low delta values. And these three are interrupted by one (orange) or two (black) residues not exhibiting helical secondary chemical shifts. Altogether it has to be made clearer that this interaction is very transient and that it is an ensemble view and that the off-rate is possibly very high (rotational tumbling is not affected as shown by the authors). The only part where this is mentioned in passing is on lines 267-270.

Thank you for your suggestion. We down-tuned the statement:

“However, TAD2 shows only a mild preference for an α -helical conformation, which appears to be slightly stabilized in presence of FOXO4^{FH} (Fig. S3a).”

Furthermore, we extended the discussion related to the dynamic nature of the interaction and the complexes by providing more analyses of the MD data (see above).

Why is the interaction of phosphorylated p53-TAD2 with FOXO4-FH and FOXO4-DRI spontaneously exothermic? (Figure S6).

As seen now in the updated table 1 the binding of both FOXO4^{FH} and FOXO4-DRI to phosphorylated TAD2 is enthalpy-driven (negative deltaH) and therefore related to a spontaneously exothermic binding event whereas in case of non-phosphorylated p53^{TAD2} the enthalpy-derived ITC signal is low and positive and binding is mostly driven by the entropic contribution. This could be explained by additional electrostatic interactions mediated by the phosphate group in phosphorylated p53 versions.

Minor concerns:

Altogether, it seems that the manuscript was prepared in a rush and it would need considerable language editing and proof-reading. A few examples:

1. Line 51: “cells cells”
2. Line 54: “including for example”.
3. Line 66: Copying should be coping
4. Line 94: FOXO4 is member, an “a” is missing and “the” afterwards.
5. Line 98: “such as for example”
6. Line 206: “vide infra”, why not just saying “see below”
7. Line 404: “what” is misplaced somehow?
8. Line 407: immense importance?
9. Line 513: affords

Panels in Figure 1 are not aligned very well (width of spectra and plots).

According to the reviewer’s comment we corrected all typos in the main text.

We tried to better align the panels in Figure 1.

REVIEWER COMMENTS

Comments are highlighted with a green background

Reviewer #1 (Remarks to the Author):

The authors have addressed essentially all the concerns I had with refined or new experiments, and the new results largely support their hypotheses.

We thank the reviewer for positive evaluation.

Reviewer #2 (Remarks to the Author):

The authors well addressed my early concerns but not much on the MD simulation part. They did provide simulation data files in the provided link. However, they didn't address my questions on the basics:

“For the MD simulations, it was not clear how extensive or how long (and how many) the simulations were conducted, whether the systems were well equilibrated (e.g., by showing RMSDs overall simulation time), or the interaction interfaces were stabilized, or say, the results shown in Fig 3 converged or not. Similar concerns apply to the FOXO4-DRI-p53 simulations.”

- 1) A basic question that should be addressed by every MD study: How long each simulation was conducted? To justify if the system is equilibrated (i.e., to be qualified to collect data), authors usually show the RMSD vs time plot at least, on which one can identify convergence of the RMSD value with time after a certain period. I'd insist this check to be a minimal requirement for MD.
- 2) Fig 3C shows RMSF and RMSD values, presumably obtained in the equilibrated state; I was trying to ask if the RMSD and RMSF values they show have been stabilized or converged (related to 1).
- 3) Another basic question is for each simulation system, how many runs were conducted? This concerns with statistics, or say, to show if ensemble samplings are well done. I understand that current study is mainly on experimental NMR characterization. However, since MD is stated as an important component, the authors should provide simple basics descriptions explicitly in the ms.

In their response, the authors stated “We have further extended the description of MD simulations”, yet without providing the update text. As I checked into the ms and SI, I could only find page 29 in main on “Molecular dynamics simulations” section, without much change at all.

We apologize for the confusion raised. Most of the requested information has already been submitted with the revised version of the manuscript. Below, we address each of the points raised.

- **Extension of the MD simulations.** We added additional info as Table S1 and specified in the Materials and Methods section the following:

(line 844) “Subsequently, each system underwent independent simulations lasting 500 ns each (totaling 1.5 μ s per system) at 298 K, employing the Langevin thermostat with a time step of 0.2 fs.”

Table S1. Details of the simulation setup.

Simulation system	Simulation box dimensions (Å^3)	Counter ions	Composition (number of protein molecules)
p53 ^{TAD2}	170356.52	7 x Na ⁺	1 molecule
p53 ^{TAD2} pT55	170356.52	8 x Na ⁺	1 molecule
p53 ^{TAD2} pS46	170356.52	8 x Na ⁺	1 molecule
p53 ^{TAD2} - FOXO4 ^{FH}	431134.80	4 x Cl ⁻	2 molecules
p53 ^{TAD2} - FOXO4-DRI	259017.37	3 x Cl ⁻	2 molecules
p53 ^{TAD2} pS46 – FOXO4 ^{FH}	431134.80	3 x Cl ⁻	2 molecules
p53 ^{TAD2} pS46 – FOXO4-DRI	259017.37	2 x Cl ⁻	2 molecules
p53 ^{TAD2} pT55 – FOXO4 ^{FH}	431134.80	2 x Cl ⁻	2 molecules
p53 ^{TAD2} pT55 – FOXO4-DRI	259017.37	3 x Cl ⁻	2 molecules
p53 ^{TAD2} pS46 pT55 – FOXO4 ^{FH}	431134.80	2 x Cl ⁻	2 molecules
p53 ^{TAD2} pS46pT55 – FOXO4-DRI	308389.67	1 x Cl ⁻	2 molecules

- **RMSD and RMSF data.** The raw data for the RMSD and RMSF values from each MD simulation can be found alongside the rest of the MD simulation data on Zenodo (see folder 3_Analysis at <https://zenodo.org/doi/10.5281/zenodo.10963887>). The RMSF values shown in Figures 3 and 5 represent the mean value per residue across all simulations. The RMSD values are presented as a distribution because, as discussed in the text, we aimed to illustrate the impact of binding on the mobility of p53^{TAD2}.

In response to the reviewer's comments, we have plotted the data to show the temporal evolution of RMSD in each MD simulation (**Figure R1**). As shown, all systems reach equilibrium within a few nanoseconds, with conformational changes observed for some systems. Considering the intrinsically disordered nature of p53^{TAD2}, we opted to perform Markov State Model (MSM) analysis.

We have included **Figure R1** as additional figure in the revised version of the manuscript (**Figure S14**).

Figure R1. Evolution of RMSD (Å) for the backbone atoms of p53TAD2, both alone and in complex with the two effectors, throughout the MD simulations.

As the systems simulated in this manuscript are highly dynamic and do not adopt a single dominant conformational state throughout the MD simulations, we refrained initially from including a plot in the manuscript for the evolution of RMSD values over time. Instead, we performed Markov State Model (MSM) analysis to better understand the conformational landscape of these systems.

Further, I noticed that the authors provided “Markov State Model” on top of MD simulation study. In their technical descriptions on page 30, again, basic descriptions are lack of: how long was each simulation, how many simulations were conducted in total, which atomic coordinates were used as metrics (distances? dihedrals? etc) or were used as input for tICA? For the MSM (Markov state model) construction, how many microstates were obtained? Why did the authors present “Reweighted free energy surface” (in Figure S8 and S8)? Since for MSM, only equilibrium simulations were conducted, no enhanced sampling nor weights are supposed to be added, so what to do with the “Reweighted”?

In the previous revision, we had included the following section:

“The conformational space of p53TAD2 was analyzed via Markov State Model (MSM).⁶¹ 1500 frames from each MD trajectory were printed with cpptraj and considered for analysis. In all cases, water molecules, counter ions, and binding partners (FOXO4-DRI, FOXO4FH) were removed before analysis. First, the number of dimensions were reduced using time-structure independent component analysis (tICA).^{62, 63} As features, we selected the backbone torsions of p53TAD2. Then, the MD simulations was divided into microstates using the clustering algorithm k-means. A lag-time of 5 ns was defined for all systems. The MSM models were validated using the Chapman-Kolmogorov test⁶⁴ with a confidence of 95%. Finally, the metastable sets were computed using the PCCA+ spectral clustering implemented in pyEMMA. As outcome, we computed the free energy surfaces and transition probabilities. The entire analysis was performed using pyEMMA⁶⁵ freely available at github (<https://github.com/markovmodel/PyEMMA>).”

In response to the reviewer's comments, we have revised the text as follows:

“The conformational space of p53TAD2 was analyzed via Markov State Model (MSM).⁶¹ 1500 frames from each MD trajectory of each simulated system (1.5 μ s per system) were printed with cpptraj and considered for analysis. In all cases, water molecules, counter ions, and binding partners (FOXO4-DRI, FOXO4FH) were removed before analysis. First, the number of dimensions were reduced using time-structure independent component analysis (tICA).^{62, 63} As features, we selected the backbone torsions of p53TAD2. Then, the MD simulations was divided into 85 microstates using the clustering algorithm k-means. A lag-time of 5 ns was defined for all systems. The MSM models were validated using the Chapman-Kolmogorov test⁶⁴ with a confidence of 95%. Finally, the metastable sets were computed using the PCCA+ spectral clustering implemented in pyEMMA. As outcome, we computed the free energy surfaces and transition probabilities. The entire analysis was performed using pyEMMA⁶⁵ freely available at github (<https://github.com/markovmodel/PyEMMA>).”

Regarding the question about the term ‘reweighted’, it is the terminology used in *pyemma* (see <https://doi.org/10.33011/livecoms.1.1.5965>,) and refers to the re-weighting the trajectory frames with stationary probabilities from the MSM (see Figures S5 and S8). It is not related to the reweighting needed in enhanced sampling techniques like metadynamics or accelerated molecular dynamics.

Indeed, MSM construction belongs to advanced MD analyses that usually demand a separate study. I'm not sure if such type of advanced analyses is really needed in current ms. It seems that the authors added this part in response to address other referees' concerns. However, from what I read in the revised ms on the MSM results (Figure S5 and S8), I have not found new insights but with technical concerns:

Page 9 (on Figure S5): As anticipated, several minima were identified, revealing that the binding of p53TAD2 to FOXO4FH alters the distribution of microstates within the two <me-lagged independent components (ICs; Fig. S5). Notably, binding to FOXO4FH reduces dispersion between IC1 and IC2, resulting in more restricted conformational changes. This suggests that p53TAD2 becomes increasingly rigid upon interaction with FOXO4. Page 14-15 (on Figure S8): Markov state modeling of p53TAD2 both in isolation and in complex with FOXO4-DRI displayed several minima, revealing that the binding of p53TAD2 to FOXO4-DRI alters the distribution of microstates within the two time-lagged independent components (Figure S8). As in case of FOXO4FH binding, binding to FOXO4-DRI reduces dispersion between IC1 and IC2, resulting in more restricted conformational changes and increasing rigidity in the bound state.

For two systems of p53TAD2 and p53TAD2-FOXO4FH shown on Figure S5, the two were separately simulated and they have different IC1 and IC2 (eigenfunctions from tICA); it seems misleading to show them on the same set of IC1-2. Similar concerns apply to Figure S8. Indeed, for any pair of A (no B bound) and A+B (B bound) systems, I figure that the same conclusions always apply: the bound system display different minima from the non-bound system. Hence, I see no points that the authors have to use MSM in current ms, although I appreciate their computational efforts, and would leave the other referee to judge if the MSM results address their concerns. In brief, I find that current ms needs to address MD basics better.

We apologize for the confusion raised. As described in the *Methods and Materials* section, '*In all cases, water molecules, counter ions, and binding partners (DRI, FOXO4, FOXO4FH) were removed before analysis*'. It means, that our MSM analysis includes **only** the backbone torsions of p53^{TAD2} **for all systems** that are discussed. As an example, in the case of p53^{TAD2} vs. p53^{TAD2}-FOXO4^{FH}, we took the backbone torsion angles of p53^{TAD2} from each system and we computed the IC1 and IC2 values at once.

We conducted 4 different MSM analysis:

- 1) p53TAD2/p53TAD2_FOXO4FH/ p53TAD2_FOXO4DRI
- 2) p53TAD2pS46/p53TAD2pS46_FOXO4FH/ p53TAD2pS46_FOXO4DRI
- 3) p53TAD2pT55/p53TAD2pT55_FOXO4FH/ p53TAD2pT55_FOXO4DRI
- 4) p53TAD2pT55S46/p53TAD2pT55S46_FOXO4FH/ p53TAD2pT55S46_FOXO4DRI

Regarding the relevance of MSM in this study, we would prefer to include the MSM in the study, also to address the points raised by the other reviewer. The complexity of the studied systems makes it challenging to analyze binding events using simple metrics such as RMSD and RMSF. Specifically, understanding the interactions between p53^{TAD2} in its different phosphorylation states and the binding to two peptides discussed in this work requires a more differentiated approach. In this context, the insights provided by MSM analysis are particularly valuable.

Reviewer #3 (Remarks to the Author):

The authors have erased most of my concerns but two new ones have come up. However, I think this should be possible to resolve in short order. Therefore, I recommend acceptance of this manuscript for publication at Nature Communications provided my two remaining concerns are satisfyingly addressed:

1. the new ITC data (mutations of hydrophobic residues compared with wild type) do not show a large enough difference that makes a single experiment convincing enough. It seems the error bars are from the fit and not from triplicates. It needs to be shown with these small difference if they are not because of experimental variation.

We performed all corresponding ITC experiments in triplicates (**Figure R2**). The results reveal that the binding affinities between the p53 1-94 mutants and FOXO4 FH are all increased when compared to wild-type p53. We replaced Figure S6 and updated table 1 accordingly.

Figure S6. p53¹⁻⁹⁴ interacts with FOXO4^{FH} via a network of hydrophobic and charge-based interactions: Titration of 25 μM of p53¹⁻⁹⁴, p53¹⁻⁹⁴I50A, p53¹⁻⁹⁴W53A, p53¹⁻⁹⁴F54A, p53¹⁻⁹⁴A3A and p53¹⁻⁹⁴ high salt with 250 μM of FOXO4^{FH} (a-f). Titration of 50 μM of p53¹⁻⁹⁴ with 500 μM of FOXO4^{DRI} in (g).

2. The explanation for the difference in affinity of ITC data and the fast exchange regime in NMR titrations, indicating weaker binding is not satisfying (different buffers). The ITC buffer actually has more salt and would thus result in a weaker affinity? Can an explanation be found in the nature of the experiments?

Figure R3 (below) illustrates the NMR titration of ^{15}N -labeled p53^{TAD2} (50 μM) with increasing concentrations of FOXO4^{FH} (50, 100, and 200 μM) in two different buffers: 20 mM HEPES pH 7.0, 50 mM NaCl, 2 mM TCEP (represented by a color gradient from red to orange), and 50 mM Na₂HPO₄/NaH₂PO₄ pH 7.5, 2 mM TCEP (represented by a color gradient from blue to magenta). The data reveal larger chemical shift perturbations of p53^{TAD2} ^1H - ^{15}N cross-peaks in the HEPES buffer compared to the phosphate-based buffer. This observation supports our hypothesis that the presence of phosphate ions in the buffer interferes with electrostatic interactions between p53 and FOXO4, leading to an apparent reduction in affinity in the phosphate buffer relative to HEPES.

Figure R3: (a) ^1H , ^{15}N HSQC spectrum of 50 μM ^{15}N -labeled p53^{TAD2} (50 μM) with increasing concentrations of FOXO4^{FH} (50, 100, and 200 μM) in two different buffers: 20 mM HEPES pH 7.0, 50 mM NaCl, 2 mM TCEP (represented by a color gradient from red to orange), and 50 mM Na₂HPO₄/NaH₂PO₄ pH 7.5, 2 mM TCEP (represented by a color gradient from blue to magenta).

The authors well addressed my early concerns but not much on the MD simulation part. They did provide simulation data files in the provided link. However, they didn't address my questions on the basics:

“For the MD simulations, it was not clear how extensive or how long (and how many) the simulations were conducted, whether the systems were well equilibrated (e.g., by showing RMSDs overall simulation time), or the interaction interfaces were stabilized, or say, the results shown in Fig 3 converged or not. Similar concerns apply to the FOXO4-DRI-p53 simulations.”

- 1) A basic question that should be addressed by every MD study: How long each simulation was conducted? To justify if the system is equilibrated (i.e., to be qualified to collect data), authors usually show the RMSD vs time plot at least, on which one can identify convergence of the RMSD value with time after a certain period. I'd insist this check to be a minimal requirement for MD.
- 2) Fig 3C shows RMSF and RMSD values, presumably obtained in the equilibrated state; I was trying to ask if the RMSD and RMSF values they show have been stabilized or converged (related to 1).
- 3) Another basic question is for each simulation system, how many runs were conducted? This concerns with statistics, or say, to show if ensemble samplings are well done. I understand that current study is mainly on experimental NMR characterization. However, since MD is stated as an important component, the authors should provide simple basics descriptions explicitly in the ms.

In their response, the authors stated “We have further extended the description of MD simulations”, yet without providing the update text. As I checked into the ms and SI, I could only find page 29 in main on “Molecular dynamics simulations” section, without much change at all.

Further, I noticed that the authors provided “Markov State Model” on top of MD simulation study. In their technical descriptions on page 30, again, basic descriptions are lack of: how long was each simulation, how many simulations were conducted in total, which atomic coordinates were used as metrics (distances? dihedrals? etc) or were used as input for tICA? For the MSM (Markov state model) construction, how many microstates were obtained? Why did the authors present “Reweighted free energy surface” (in Figure S8 and S8)? Since for MSM, only equilibrium simulations were conducted, no enhanced sampling nor weights are supposed to be added, so what to do with the “Reweighted”?

Indeed, MSM construction belongs to advanced MD analyses that usually demand a separate study. I'm not sure if such type of advanced analyses is really needed in current ms. It seems that the authors added this part in response to address other referees' concerns. However, from what I read in the revised ms on the MSM results (Figure S5 and S8), I have not found new insights but with technical concerns:

Page 9 (on Figure S5): As anticipated, several minima were identified, revealing that the binding of p53TAD2 to FOXO4FH alters the distribution of microstates within the two time-lagged independent components (ICs; Fig. S5). Notably, binding to FOXO4FH reduces dispersion between IC1 and IC2, resulting in more restricted conformational changes. This suggests that p53TAD2 becomes increasingly rigid upon interaction with FOXO4.

Page 14-15 (on Figure S8): Markov state modeling of p53TAD2 both in isolation and in complex with FOXO4-DRI displayed several minima, revealing that the binding of p53TAD2 to FOXO4-DRI alters the distribution of microstates within the two time-lagged independent components (Figure S8). As in case of FOXO4FH binding, binding to FOXO4-DRI reduces dispersion between IC1 and IC2, resulting in more restricted conformational changes and increasing rigidity in the bound state.

For two systems of p53TAD2 and p53TAD2-FOXO4FH shown on Figure S5, the two were separately simulated and they have different IC1 and IC2 (eigenfunctions from tICA); it seems misleading to show them on the same set of IC1-2. Similar concerns apply to Figure S8. Indeed, for any pair of A (no B bound) and A+B (B bound) systems, I figure that the same conclusions always apply: the bound system display different minima from the non-bound system. Hence, I see no points that the authors have to use MSM in current ms, although I appreciate their computational efforts, and would leave the other referee to judge if the MSM results address their concerns. In brief, I find that current ms needs to address MD basics better.